# Regulatory mechanism of cold-inducible diapause in *Caenorhabditis elegans*

Makoto Horikawa [1] ✉, Masamitsu Fukuyama [2], Adam Antebi [3,4] & Masaki Mizunuma [1,5] ✉

Temperature is a critical environmental cue that controls the development and lifespan of many animal species; however, mechanisms underlying low-temperature adaptation are poorly understood. Here, we describe cold-inducible diapause (CID), another type of diapause induced by low temperatures in *Caenorhabditis elegans*. A premature stop codon in heat shock factor 1 (*hsf-1*) triggers entry into CID at 9 °C, whereas wild-type animals enter CID at 4 °C. Furthermore, both wild-type and *hsf-1(sy441)* mutant animals undergoing CID can survive for weeks, and resume growth at 20 °C. Using epistasis analysis, we demonstrate that neural signalling pathways, namely tyraminergic and neuromedin U signalling, regulate entry into CID of the *hsf-1* mutant. Overexpression of anti-ageing genes, such as *hsf-1*, XBP1/*xbp-1*, FOXO/*daf-16*, Nrf2/*skn-1*, and TFEB/*hlh-30*, also inhibits CID entry of the *hsf-1* mutant. Based on these findings, we hypothesise that regulators of the *hsf-1* mutant CID may impact longevity, and successfully isolate 16 long-lived mutants among 49 non-CID mutants via genetic screening. Furthermore, we demonstrate that the nonsense mutation of MED23/*sur-2* prevents CID entry of the *hsf-1(sy441)* mutant and extends lifespan. Thus, CID is a powerful model to investigate neural networks involving cold acclimation and to explore new ageing mechanisms.

Temperature is one of the most critical environmental cues that impact development, reproduction, behaviour, and lifespan in many animal species. In ectotherms, lifespan is generally reduced with increasing temperature, and the developmental span is accelerated in the range of habitable temperatures[1–4]. This simple correlation is conventionally thought to be due to the rate of metabolism, namely chemical reactions at different temperatures. In addition to temperature shifts affecting developmental speed and lifespan in *Caenorhabditis elegans*, recent studies have demonstrated that specific regulatory mechanisms control development, survival, and lifespan in response to cold temperatures[2,5–8].

Several neuroscience studies have revealed the main thermosensory neurons, AFD, AWC, and ASI, and the neural circuits governing thermotaxis behaviour[9–11]. In addition, more than 10 neurons have been identified with thermosensory modality and are partially shared with other sensory machinery, including chemosensory, osmosensory, and mechanosensory neurons. Some neurons are notably involved in the cold response[7,12–14]. In the context of cold adaptation at 2 °C, ASJ, ADL and ASG neurons have roles in cold adaptation[7,15,16]. A *C. elegans* homologue of a mammalian cold sensor molecule TRPA1, *trpa-1*, and its downstream protein kinase, *pkc-2*, are not only involved in the effects of cold on behaviour but also in the regulation of lifespan at low

[1]Unit of Biotechnology, Graduate School of Integrated Sciences for Life, Hiroshima University, Higashi-Hiroshima, Japan. [2]Laboratory of Physiological Chemistry, Graduate School of Pharmaceutical Sciences, University of Tokyo, Tokyo, Japan. [3]Max Planck Institute for Biology of Ageing, Cologne, Germany. [4]Cologne Excellence Cluster on Cellular Stress Responses in Aging-Associated Diseases (CECAD), University of Cologne, Cologne, Germany. [5]Hiroshima Research Center for Healthy Aging (HiHA), Hiroshima University, Higashi-Hiroshima, Japan. ✉e-mail: mhorikawa0204@hiroshima-u.ac.jp; mmizu49120@hiroshima-u.ac.jp

temperatures through insulin signalling[4,8]. These findings reveal that neural circuits play important roles in a wide range of cold responses that affect behaviour, survival, and lifespan.

Other mechanisms also reportedly regulate thermal adaptation. Fatty acid metabolism modulates membrane fluidity in response to temperature and contributes to thermal adaptation from bacterial to mammalian cells[17,18]. Homologues of nuclear hormone receptor HNF4, *nhr-49*, mediator 15, *mdt-15*, and adiponectin receptor, *paqr-2*, have been demonstrated to control development, fertility, and lifespan in response to cold temperatures via the regulation of fatty acid metabolism[6,19,20]. Chaperone proteins also play key roles in thermal responses, such as heat adaptation and cold tolerance, in several eukaryotic species[21–23]. We previously discovered that loss-of-function mutations in *daf-41*, a *C. elegans* homologue of co-chaperone p23, leads to a reduced lifespan at 15 °C but to an extended lifespan at 25 °C. Moreover, the gain-of-function mutation of *hsp-90*, a partner protein of p23, increases lifespan only in response to low temperatures[5]. Although many studies have been conducted on the effects of cold environments in *C. elegans*, its cold response mechanism remains poorly understood, especially the regulatory mechanism of longevity at cold temperatures. One of the challenges of this research area is the absence of reporter strains that reflect cold environments and/or cold-inducible phenotypes that can be used as tools to explore regulatory mechanisms of cold adaptation and ageing.

In *C. elegans* ageing research, the dauer diapause phenotype has been widely used as a biological proxy for the isolation of long-lived mutants, since it is an easily scored phenotype compared to directly measuring lifespan[24–26]. Dauer diapause is induced by unfavourable environmental cues, such as starvation, high population density and high temperatures, and entails interactions among multiple longevity mechanisms, including insulin, mechanistic target of rapamycin (mTOR), chaperone, transforming growth factor (TGF-β), and steroidal signalling pathways. Notably, *daf-2* mutants exhibit dauer-constitutive as well as remarkably long-lived phenotypes. Other diapause states, such as L1 diapause and adult reproductive diapause (ARD), are triggered by starvation and also regulated by overlapping longevity mechanisms[27–31]. These diapause states are also regulated by neural circuits involved not only in nutrient response but also in chemosensory and thermosensory pathways[32]. However, studies focusing on whether such diapause states are induced at low temperatures remain scarce.

In this study, we discovered the diapause state induced by cold stimuli and associated with longevity. Thereafter, we used the phenotype as a tool to investigate the regulatory mechanisms of ageing in response to low temperatures. We found that a premature stop codon (PTC, the *sy441* allele) in *hsf-1*, a transcription factor mediating the heat shock response (HSR), displayed reversible developmental arrest at 9 °C, and in N2 (wild-type) animals at 4 °C. We termed this type of arrested state cold-inducible diapause (CID). We discovered that *hsf-1(sy441)* mutant CID is controlled by a neural network and several longevity mechanisms. Furthermore, we demonstrated that the CID phenotype of the *hsf-1(sy441)* mutant can be leveraged for genetic screening of long-lived mutants, and identified *sur-2*/MED23 as a regulator of both CID entry and lifespan. Overall, we established that the *hsf-1(sy441)* mutant CID can be used to unravel the complex interaction between the thermosensory system in neural circuits, the regulatory mechanism of development and that of longevity in response to cold temperatures.

## Results
### Chaperone proteins play a role in cold adaptation
Chaperone proteins play a key role in thermal responses to both heat and cold tolerance in several eukaryotic species[22,23,33]; however, the function of the heat shock response (HSR) during cold acclimation in *C. elegans* is unclear. To elucidate the role of chaperone proteins at low

temperatures, we first determined the lower temperature threshold for growth in wild-type (N2) animals. We discovered that wild-type worms could develop into reproductive adults at 10 °C (Fig. 1a). Although the generation time at 10 °C was approximately three-fold longer than that at 15 °C, wild-type animals lived 1.5 times longer than those at 15 °C (Fig. 1a). By narrowing the lowest threshold temperature, we found that wild-type worms were able to develop and reproduce at 9 °C (Fig. 1b) but ceased growth at 4 °C (Fig. 2a). Next, we evaluated cold-inducible phenotypes of various chaperones and their regulator mutants at 9 °C and found that *hsf-1(sy441)* mutants ceased development at the L1 stage, even in the presence of bacterial food (Fig. 1b, c). Mutations in orthologues of ER resident heat shock protein (HSP) 70, *hsp-3* and *hsp-4*, led to delayed development and 100% sterility (Supplementary Fig. 1a, b). Their gene expression is regulated by the IRE1-XBP1 pathway, which is activated upon endoplasmic reticulum (ER) stress[34]. Accordingly, the growth of *ire-1(ok799)* mutants was slower than that of wild-type animals at 9 °C (Supplementary Fig. 1a). The development of other ER stress response pathway mutants[34], such as *atf-6(ok551)*, *pek-1(ok275)*, and *xbp-1(zc12)*, was similar to that of wild-type animals, even at 9 °C (Supplementary Fig. 1a, Supplementary Table 1). Additionally, *daf-41(ok3052)*, *hsp-90(p673)*, and other HSP mutants developed normally, similar to wild-type worms at 9 °C (Supplementary Fig. 1a, Supplemental Table 1). We also analysed a number of mutants that were previously reported as cold sensitive for development and longevity phenotypes. We found that both *pkc-2(ok563)* and *paqr-2(tm3410)* mutants developed slowly at 9 °C, and that the *paqr-2(tm3410)* mutant exhibited a sterile phenotype (Supplementary Fig. 1a, b)[4,6–8]. However, other cold-sensitive and ageing mutants (Supplemental Table 1) did not exhibit cold sensitivity. These results suggest that the *hsf-1*, *ire-1*, and ER-resident HSP70 orthologues play an important role in cold acclimation.

Based on these results, we further focused on the cold-sensitive phenotypes of the *hsf-1* mutants. We observed that the *hsf-1(sy441)* worms arrested development in the presence of bacterial food at 9 °C, with morphological features similar to starvation-induced L1 arrest animals[29,31] (Fig. 1d, Supplementary Fig. 2a, b). With time, the number of L1 arrested animals was reduced, and that of moulted and dead worms increased, but the average body size was unchanged (Supplementary Fig. 2c, d). In contrast to cold-induced developmental arrest, the survival span of *hsf-1(sy441)* mutants in the state of L1 arrest was much shorter than that of wild-type animals (Fig. 1e). In cold exposure experiments, we found that a number of *hsf-1(sy441)* worms became mobile when they were shifted from 9 °C to room temperature even at day 20 (∼ 22 °C). These observations suggest that the *hsf-1* mutants exhibit temporally arrested development in response to cold environments. Subsequently, we performed a recovery experiment with the arrested *hsf-1(sy441)* worms that were incubated at 9 °C and transferred them to a permissive temperature (20 °C) at 20, 40, and 60 days after cold-induced developmental arrest. The *hsf-1* mutants that recovered on day 20 grew and became reproductive adults (Fig. 1f, g), although the developmental speed of *hsf-1* mutants was slightly slower than that of worms that did not undergo cold-induced developmental arrest. In addition, *hsf-1(sy441)* worms that were removed from cold exposure at 40 and 60 days only produced a few reproductive adults (Fig. 1f, g). We observed that developmental speed was significantly delayed in the recovery phase with an increasing period of arrest. Previous studies have reported that developmental temperature affects cold acclimation at 2 °C[15]. Therefore, we also examined the effect of maternal temperature on cold-induced developmental arrest and found that the maternal condition did not alter the development of wild-type and *hsf-1(sy441)* mutants (Supplementary Fig. 2e). These results suggest that the cold-induced developmental arrest of *hsf-1* mutants is another type of diapause induced by cold temperatures (9 °C) even in the presence of food, which is reversible at elevated temperatures.

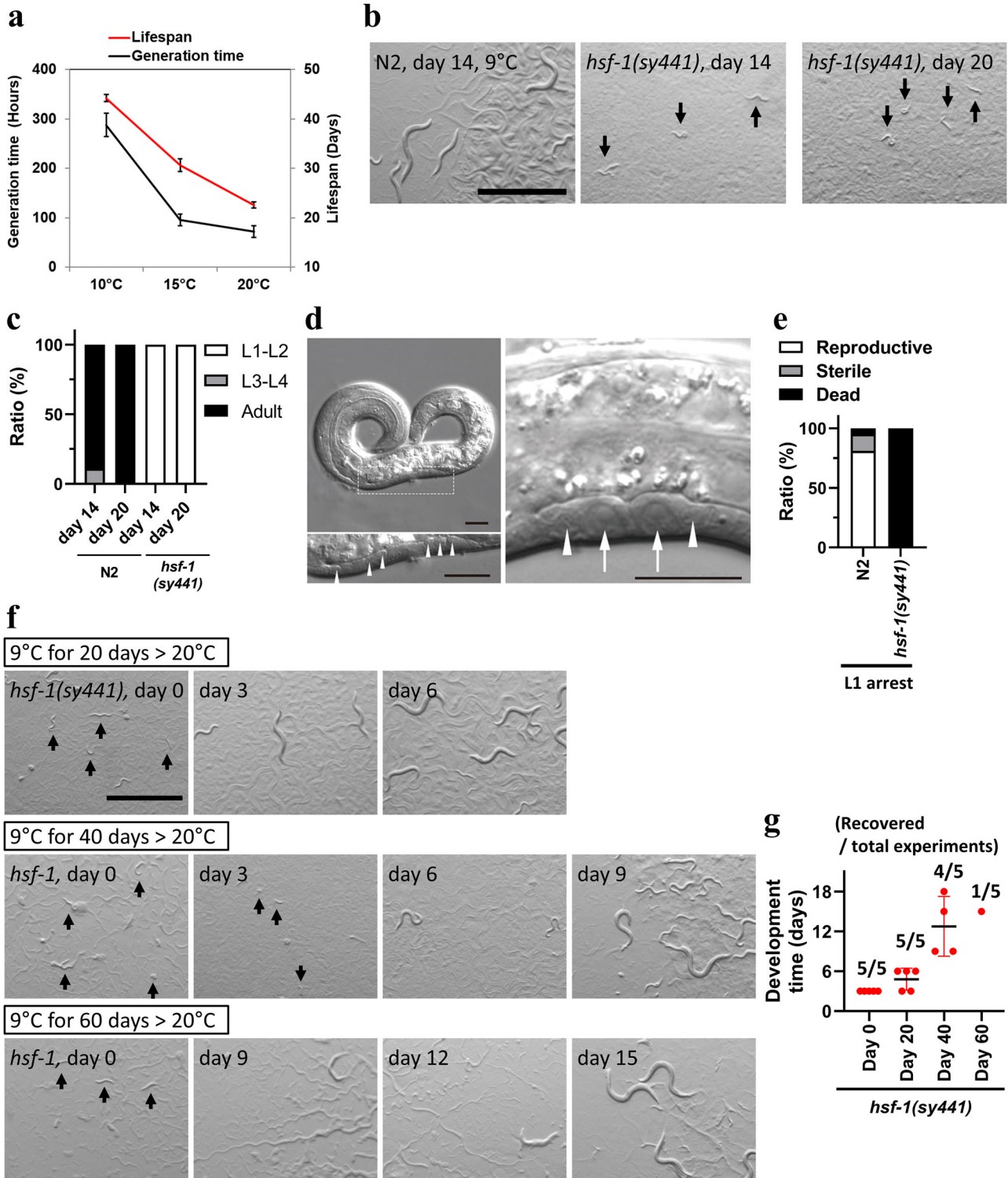

**Developmental arrest is induced in wild-type animals at 4 °C**
Next, we examined whether developmental arrest is induced in wild-type animals at temperatures below 9 °C. We exposed eggs to 4 °C for 14 days and observed that no eggs hatched during this period (Fig. 2a). Furthermore, chilled embryos did not recover after transferring them to 20 °C. We also exposed early L1 larvae (EL1) to a temperature of 4 °C at the time when the developmental arrest was induced in *hsf-1(sy441)* mutants at 9 °C. Wild-type EL1 animals did not develop but were able to survive at 4 °C for 14 days, and successfully developed into reproductive adults after transfer to a temperature of 20 °C (Fig. 2b). The

*hsf-1* mutants also exhibited a diapause phenotype at 4 °C, similar to 9 °C, but displayed morphological abnormalities. They were typically sterile, exhibited a protruding vulva (Pvl), and were egg laying defective (Egl) after recovery from the diapause state at 20 °C (Fig. 2c). Moreover, the recovery speed of the *hsf-1* mutants from diapause at 4 °C was significantly slower when compared to wild-type animals or *hsf-1(sy441)* animals that were not treated with cold exposure (Figs. 1g, 2d). These results demonstrate that a severe cold temperature (4 °C) induces diapause even in wild-type animals. The *hsf-1* mutant exhibited a more severe phenotype at 4 °C than at 9 °C. We propose that the *hsf-1*

**Fig. 1 | *hsf*-1 triggers entry into cold-induced developmental arrest. a** The lifespan of wild-type animals increased linearly due to a reduction in temperature, whereas the development speed decreased drastically between 10 °C and 15 °C. Red line indicates The bars indicate the mean ± standard error of the mean (SEM). Red line indicates mean lifespan and black line indicates mean generation time. Ageing experiments performed in 3 replicates with 120 animals and developmental analysis performed in 3 replicates with more than 100 worms at each temperature. **b**, **c** Cold environment (9 °C)-induced growth arrest in *hsf-1(sy441)* mutant animals at the L1 stage. **c** Graph representing the development profile. White indicates the relative worm population of L1-L2 larvae, grey that of L3-L4 larvae and black that of adult worms. **d** Microscopy images of the arrested *hsf-1(sy441)* mutant worms. An L1-arrested animal on day 21 (left panels). No P cells were detected around the bodies of neuronal cells (arrowheads) along the ventral midline, indicating that the migration of P cells was not initiated in this animal. (right) Migrated P cells (arrows) near neuronal cells (arrowheads) along the ventral nerve cord in an animal arrested on day 7. **e** The *hsf-1(sy441)* mutant worms were sensitive to starvation. L1 arrest was induced without a bacterial diet and recovered with a bacterial diet on day 5. White indicates the relative worm population of reproductive adults, grey that of sterile adults and black that of dead larvae. **f**, **g** *hsf-1(sy441)* mutant worms have the potential to resume development in response to a temperature shift from 9 °C to 20 °C after 20, 40, and 60 days of cold exposure. **g** Recovery experiments were repeated five times with more than 100 animals. Each dot indicates the day reproductive adults appeared on plates. The bars indicate the mean ± standard deviation (SD). $n \geq 3$ biological replicates with more than 50 animals (**c**, **e**). Graphs represent one of the biological replicates (**c**, **e**, **g**). Scale bars, 1 mm (**b**, **f**) and 10 μm (**d**). Black arrows indicate arrested worms (**b**, **f**). Source data are provided as a Source Data file.

gene governs the temperature threshold that triggers worms to enter diapause in response to cold temperatures. We term this type of diapause phenotype induced by cold exposure, Cold-inducible diapause (CID).

## Interaction between developmental timing and CID

Several studies have reported that *C. elegans* enters into diapause at different developmental stages; early L1 (L1 arrest), L1-L2 transition (dauer) and L4-adult transition (adult reproductive diapause)[30,31,35]. Therefore, we performed cold exposure experiments at four larval stages (L1-L4) to investigate the time specificity of CID in wild-type and *hsf-1(sy441)* worms at 4 °C and 9 °C.

We observed that with a cold shift to 9 °C in the embryo, development was arrested in *hsf-1(sy441)* mutant animals at the L2-larval stage. A cold shift in later larval stages (L3–L4) resulted in development to reproductive adults (Fig. 3a, b). When *hsf-1* mutants were shifted to 9 °C after the L3 stage, a small number of larvae grew into adults and laid eggs. These results indicate that cold stimuli (9 °C) induce developmental arrest in the presence of food and at different developmental times relative to starvation-induced L1 arrest[29,31,36]. At 4 °C, wild-type and *hsf-1(sy441)* worms survived for 14 days with a cold shift in L1 to L4 larvae (Fig. 3c, d, and Supplementary Fig. 3). This result indicates that CID is induced independently of developmental stage. However, we also found that the survival rate of L1 and L4 larvae at 4 °C was slightly higher than that of the L2 and L3 larvae (Fig. 3d). This suggests that different developmental stages have different capacities to tolerate cold temperatures.

## Neural circuits are involved in CID entry of the hsf-1(sy441) mutant

Next, we investigated the regulatory mechanism of CID using *hsf-1(sy441)* mutants at 9°C, where we expect to observe a rescue phenotype from CID. Neural circuits have been reported to trigger dauer entry and L1 arrest[24,36–38]. Therefore, we investigated the role of neural circuits in regulating CID entry. CID entry of the *hsf-1* mutant was suppressed by *unc-13* and *unc-31* mutations (Fig. 4a–c). *unc-13* is involved in neurotransmitter release, while *unc-31* controls neuropeptide secretion[39,40]. Thus, CID entry of the *hsf-1* mutant is regulated by both neurotransmitter and neuropeptide signalling. Additionally, we defined as non-CID all mutants carrying the *hsf-1(sy441)* mutation that are able to develop at 9°C, even if the strains exhibit a different developmental phenotype from the wild-type (such as the *hsf-1; unc-13* mutant).

We then examined the function of insulin signalling, which controls L1 arrest, dauer diapause, and longevity[26,29,31]. The *C. elegans* genome encodes 40 insulin-like peptides, but only one orthologue of the insulin receptor, *daf-2*[41]; therefore, we investigated the function of *daf-2* in the regulation of CID of the *hsf-1* mutant. We found that *daf-2(e1370)* animals did not enter into CID at 9 °C and the *daf-2* mutation did not rescue CID entry of *hsf-1* mutants (Supplementary Fig. 4a). This result suggests that insulin-like peptides do not trigger *hsf-1* mutant CID. Therefore, we investigated neuropeptides that trigger dormancy in other model organisms, focusing on cold-induced reproductive dormancy in *Drosophila*. Reproductive dormancy is induced at low temperatures and short photoperiods, and hugin, a neuropeptide and functional homologue of mammalian neuromedin U, projects to the corpus allatum, which is involved in the regulation of reproductive dormancy[42–44]. The *C. elegans* genome does not contain an orthologue of *Drosophila* hugin, but does contain an orthologue of neuromedin U, *capa-1*, and its receptor genes, *nmur-1, nmur-2, and nmur-3*[45,46]. Therefore, we investigated the effects of *capa-1* and *nmur-1* in CID formation and found that mutations in the neuromedin U peptide *capa-1* and its receptor *nmur-1* suppress CID entry of the *hsf-1* mutant (Fig. 4d–f, Supplementary Fig. 4b-c). Additionally, *capa-1(ok3065)* and *hsf-1; capa-1* mutants developed more slowly than the wild-types. Previous studies have reported that *capa-1*/neuromedin U signalling is involved in salt sensation and longevity in response to food conditions[45,46]. Our finding indicates that neuromedin U signalling is also associated with cold sensory machinery. Further, we investigated the role of monoamine neurotransmitters in the entry into *hsf-1* mutant CID. The genes *bas-1, cat-2, tbh-1, tdc-1*, and *tph-1* encode biosynthesis enzymes for monoamine neurotransmitters in *C. elegans* (Supplementary Fig. 4d referring to Chase and Koelle, 2007[47]). Our results indicate that the *tdc-1* mutation alone rescued the CID entry of the *hsf-1* mutant (Fig. 4g–i), while other monoamine biosynthesis genes had no roles in *hsf-1* mutant CID at 9 °C (Supplementary Fig. 4f–g). Although the developmental speed of *tdc-1* and *hsf-1; tdc-1* mutants is almost the same as that of wild-type animals at 20 °C, *hsf-1; tdc-1* mutants exhibited a slow growth phenotype at 9 °C (Fig. 4g-i, Supplementary Fig. 4e, h). In sum, these findings demonstrate that neuromedin U and tyramine are critical signalling molecules involved in *hsf-1* mutant CID, and CID regulatory mechanisms are distinct from dauer diapause and L1 arrest[24,29]. *Capa-1* and *tdc-1* genes are also involved in development at low temperatures.

## Functional analysis of HSF-1 in regulation of CID of the hsf-1(sy441) mutant

Next, we investigated how and where *hsf-1* regulates CID entry at 9 °C. We first confirmed whether *hsf-1* controls CID by reduction of its function through rescue experiments and found indeed that transgenic overexpression of *hsf-1* prevents CID in the *hsf-1* mutants (Fig. 5a, b). This supports the idea that CID of the *hsf-1* mutant is trigged by the protein truncation of the *hsf-1(sy441)* mutation. The HSF-1 protein forms a trimer in response to heat stress, and thereby acts as a transcription factor to upregulate gene expressions of HSPs[48,49]. Therefore, we examined the role of HSF-1 trimerisation in the regulation of *hsf-1* mutant CID. We confirmed that single copy HSF-1::GFP rescued the developmental arrest phenotype of *hsf-1* mutants at 25 °C as previously reported[49]; however, HSF-1(R145A)::GFP carrying a point mutation on the DNA binding domain could not restore it (Fig. 5d, f).

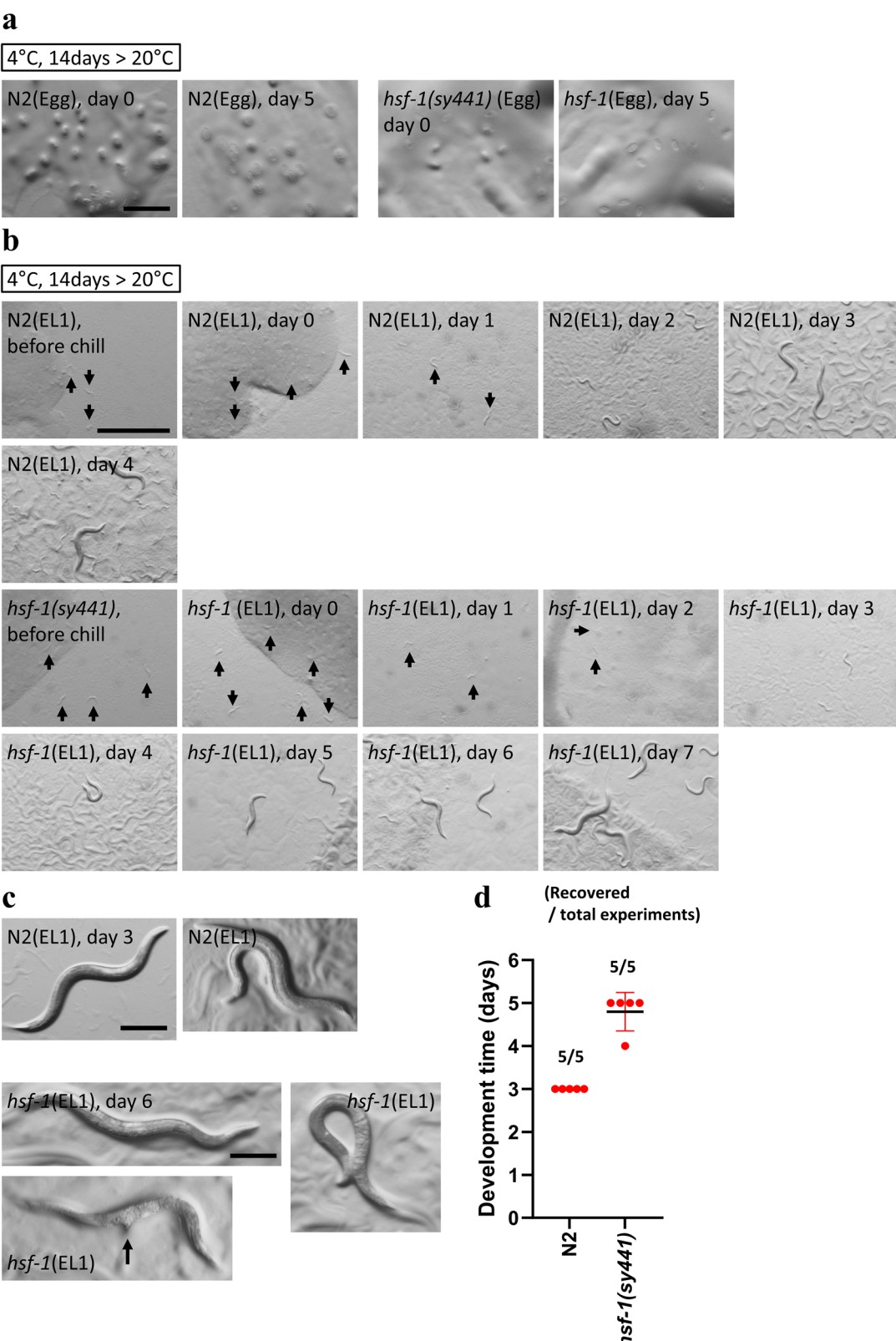

**Fig. 2 | Wild-type animals enter into a diapause state at 4 °C. a** Eggs were unhatched at 4 °C and not rescued by warming to 20 °C after cold exposure. **b–d** Cold treatment at 4 °C induced diapause in wild-type animals and *hsf-1(sy441)* mutants. *hsf-1(sy441)* mutants exhibited delayed development and morphological abnormalities after recovery from diapause induced at 4 °C. **d** Recovery experiments were repeated five times with more than 100 animals. Each dot indicates the day reproductive adults appeared on plates. The bars indicate the mean ± standard deviation (SD). *n* ≥ 3 biological replicates with more than 50 embryos and animals (**a**–**c**). Scale bars, 200 μm (**a**, **c**) and 1 mm (**b**). Arrows indicate L1 larvae and arrested worms (**b**), and protruding vulva (**c**). Source data are provided as a Source Data file.

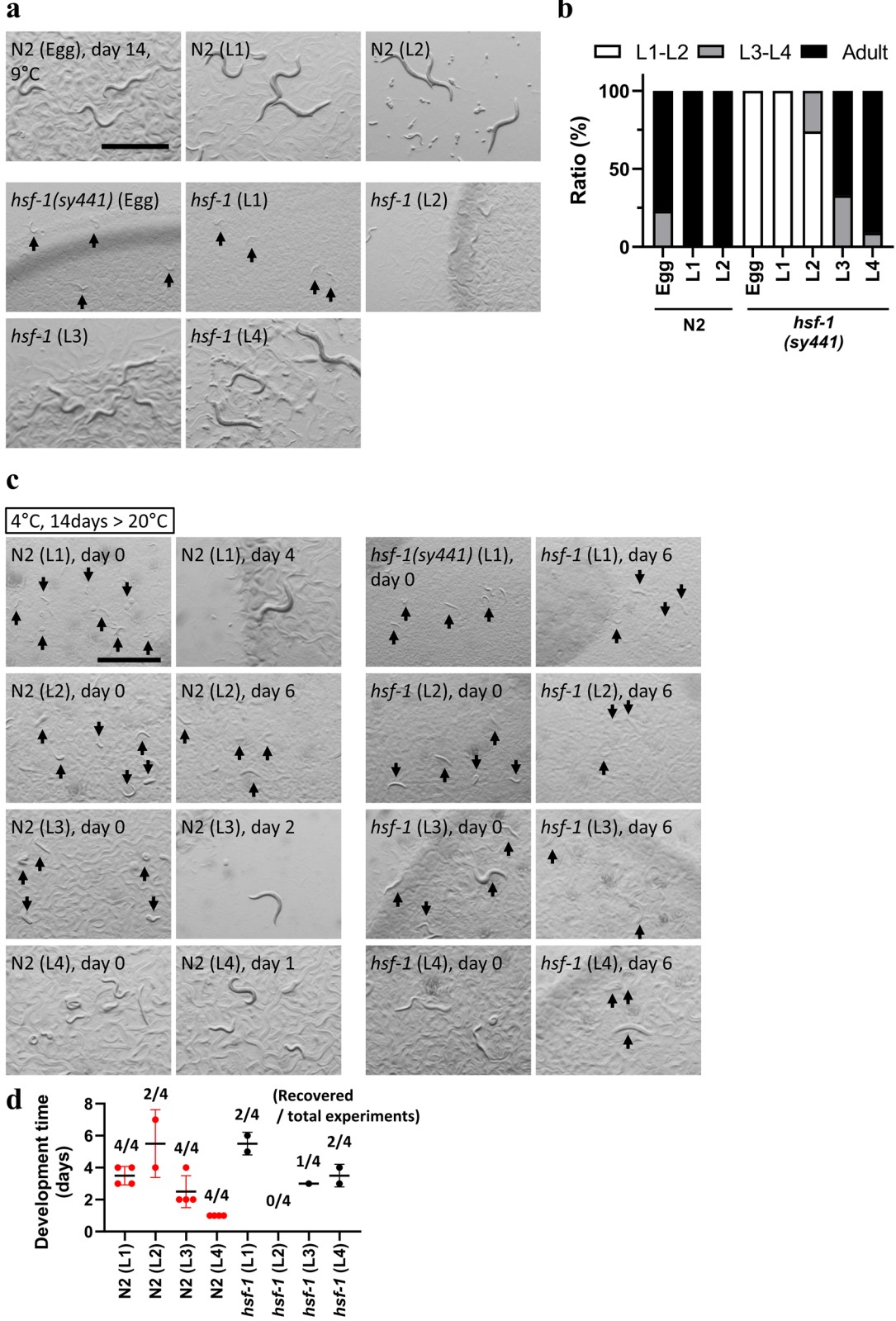

**Fig. 3 | Cold-inducible diapause (CID) is induced at several developmental stages. a, b** Developmental arrest was observed with a cold shift at 9 °C from the egg, L1, and L2 stages, but reproductive adult worms stochastically appeared from L3 and L4 worms in *hsf-1(sy441)* mutants. White indicates the relative worm population of L1-L2 larvae, grey that of L3-L4 larvae and black that of adult worms. **c, d** CID was induced by a cold shift at 4 °C from L1, L2, L3 and L4 stages in wild-type animals and *hsf-1(sy441)* mutants, but the survival ratio decreased at the L2 stage in both strains. **d** Recovery experiments were repeated four times with more than 100 animals. Each dot indicates the day reproductive adults appeared on plates. Red dots indicate wild-type animals, black dots *hsf-1(sy441)* mutants. The bars indicate the mean ± standard deviation (SD). $n \geq 3$ biological replicates with more than 50 animals (**a**–**c**). Graphs represent one of the biological replicates (**b**). Scale bars, 1 mm (**a, c**). Arrows indicate small arrested and dead worms (**a, c**). Source data are provided as a Source Data file.

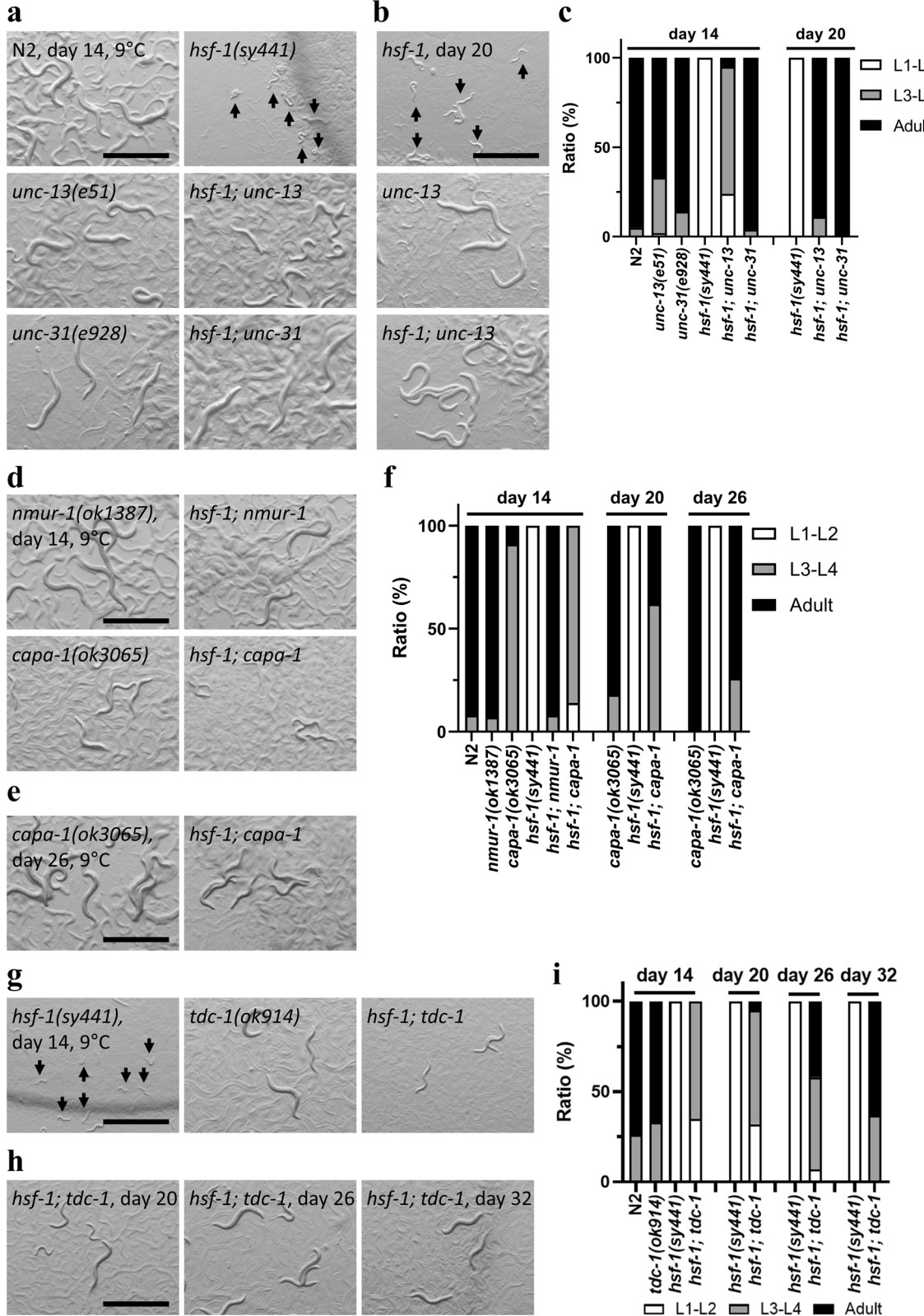

**Fig. 4 | The tyraminergic and neuromedin U signalling pathways are involved in the regulation of *hsf-1(sy441)* mutant CID.** **a**–**c** Mutations affecting the release of neurotransmitters *(unc-13)* and neuropeptides *(unc-31)* suppressed CID entry at 9 °C in the *hsf-1(sy441)* mutants. **d**–**f** Nematode homologue of the neuromedin U peptide and its receptor-regulated CID entry. (**g**–**i**) A mutation in tyrosine decarboxylase that synthesises tyramine from tyrosine inhibited CID entry. Images were obtained on days 14 (**a**, **d**, **g**), 20 (**b**, **h**-[left]), 26 (**e**, **h**-[middle]), and 32 (**h**-[right]). *n* ≥ 3 biological replicates with more than 50 animals (**a**–**i**). Graphs represent one of the biological replicates. White indicates the relative worm population of L1–L2 larvae, grey that of L3–L4 larvae and black that of adult worms (**c**, **f**, **i**). Scale bars, 1 mm (**a**, **b**, **d**, **e**, **g**, **h**). Arrows indicate arrested worms (**a**, **b**, **g**). Source data are provided as a Source Data file.

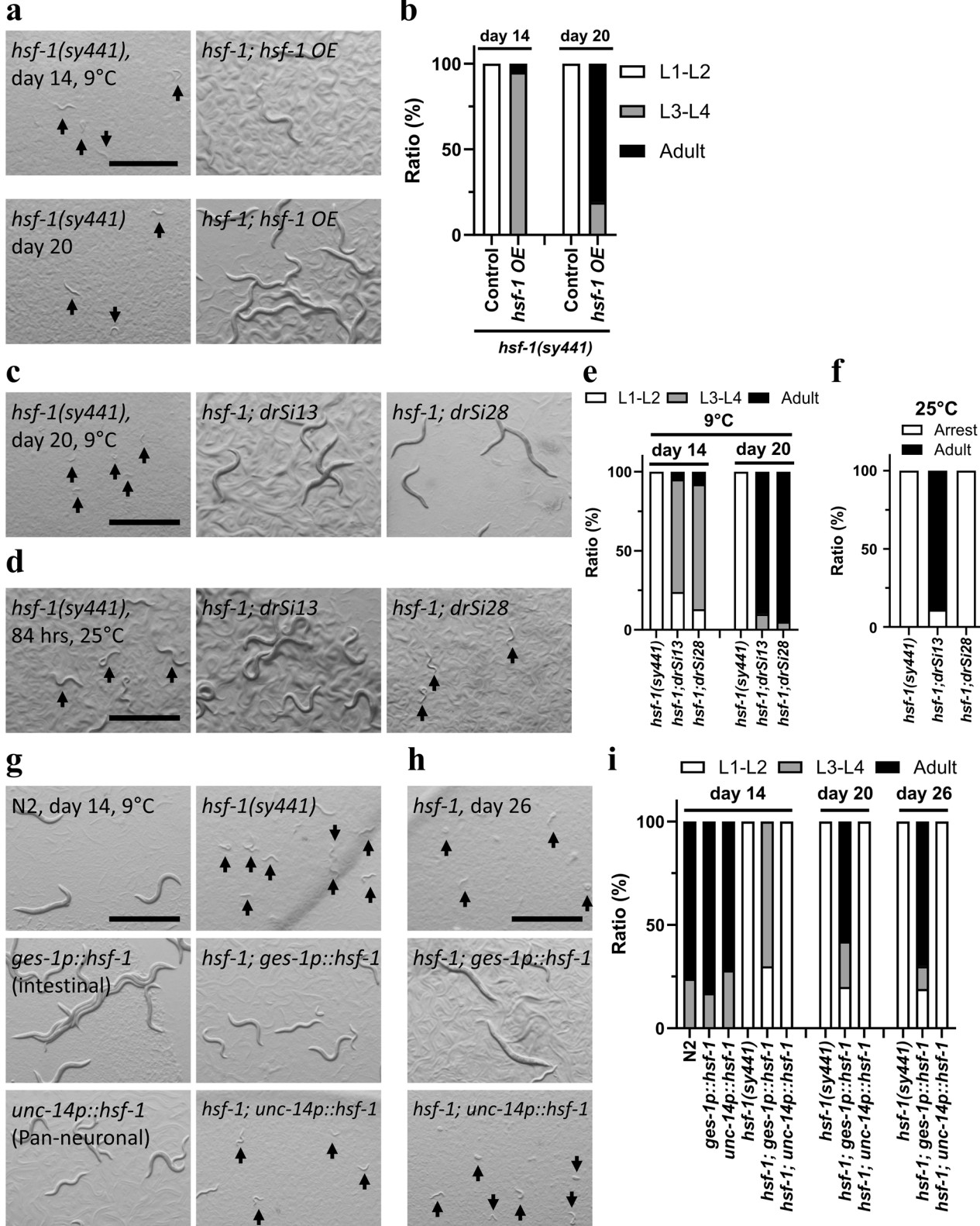

**Fig. 5 | HSF-1 is involved in CID formation of *hsf-1(sy441)* mutant independently of the heat-shock response. a**, **b** Overexpression of *hsf-1* inhibited entry of *hsf-1(sy441)* mutants into CID. **c**–**f** *hsf-1; drSi13[hsf-1::gfp]* suppressed CID entry at 9 °C and rescued developmental arrest at 25 °C. *hsf-1; drSi28[hsf-1(R145A)::gfp]* inhibited CID but did not restore developmental arrest. **g**–**i** Genetic rescue of *hsf-1* using an intestine-specific *ges-1* promoter inhibited CID entry. The neuron-specific *unc-14* promoter had no role in the process. Images were obtained at 84 h (**d**), and days 14

(**a**, **g**), 20 (**a**, **c**), and 26 (**h**). *n* ≥ 3 biological replicates with more than 50 animals (**a**–**i**). Graphs represent one of the biological replicates. White indicates the relative worm population of L1-L2 larvae, grey that of L3-L4 larvae and black that of adult worms (**b**, **e**, **i**). White indicates relative population of developmentally arrested larvae, black that of adult worms (**f**). Scale bars, 1 mm (**a**, **c**–**d**, **g**–**h**). Arrows indicate small arrested worms (**a**, **c**, **d**, **g**, **h**). Source data are provided as a Source Data file.

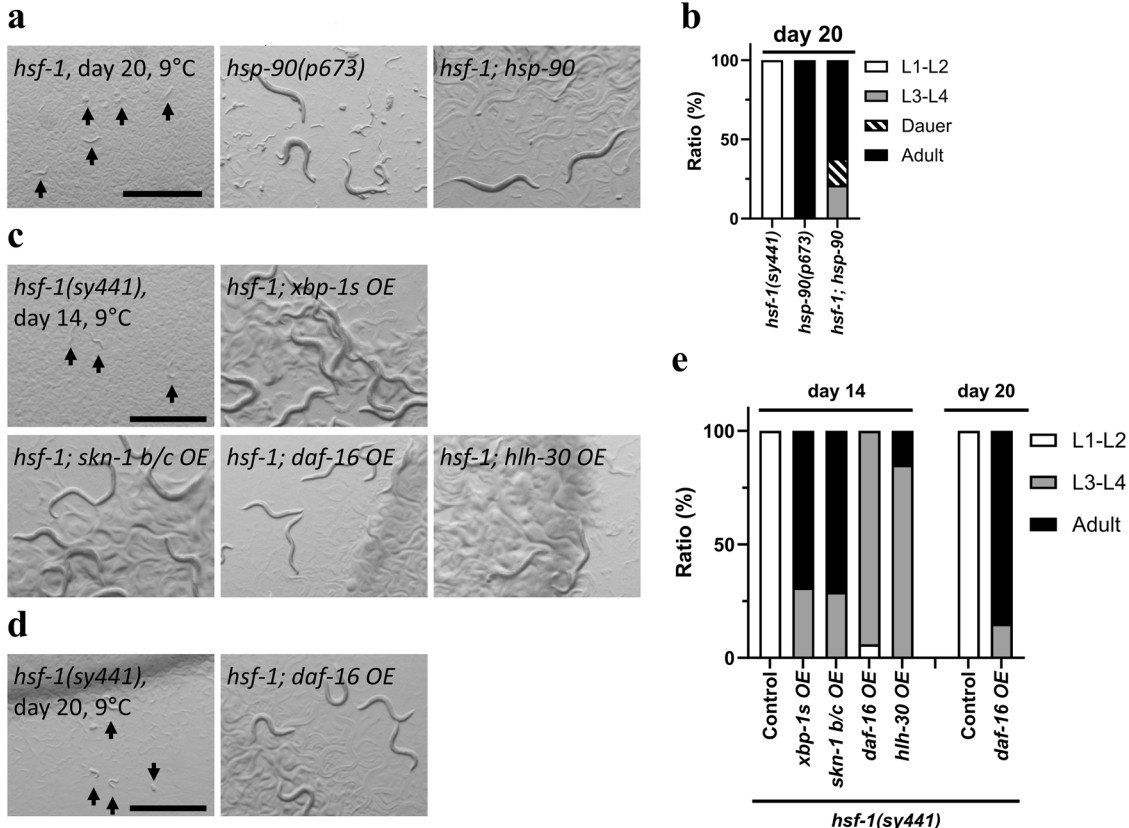

**Fig. 6 | Anti-ageing genes involved in CID entry of *hsf-1(sy441)* mutant. a, b** A gain-of-function mutation of *hsp-90* blocked CID induction in *hsf-1(sy441)* mutants. **c**–**e** Overexpression of anti-ageing genes, *xbp-1s* (active form), *skn-1 b/c*, *hlh-30*, and *daf-16*, inhibited CID induction in *hsf-1(sy441)* mutants. Scale bars, 1 mm. Images were obtained on days 14 (**c**) and 20 (**a, d**). *n* ≥ 3 biological replicates with more than 50 animals (**a**–**e**). Graphs represent one of the biological replicates. White indicates the relative population of L1-L2 larvae, grey that of L3-L4 larvae, diagonal lines that of dauer and black that of adult worms. (**b, e**). Arrows indicate arrested worms (**a, c, d**). Source data are provided as a Source Data file.

Surprisingly, we found that both HSF-1::GFP and HSF-1(R145A)::GFP rescued the CID entry of the *hsf-1* mutant at 9 °C (Fig. 5c, e). HSF-1::GFP and HSF-1(R145A)::GFP did not induce CID in wild-type animals (Supplementary Fig. 5a–c). This result suggests that the transcriptional activity of HSF-1 is not required for *hsf-1* mutant CID. The *hsf-1(sy441)* stop codon is located near the C-terminus of the HSF-1 and truncates the protein before the transactivation domain[50]. By inference, the transactivation domain of HSF-1 has an important role in cold acclimation. We further tested whether *hsf-1* plays a role in CID control using tissue-specific gene expression strains[51]. Although the neurotransmitter and neuropeptide pathways regulate CID entry in the *hsf-1* mutant, neural expression of *hsf-1* (*unc-14p::hsf-1*) did not rescue CID entry, but intestinal expression of *hsf-1* (*ges-1p::hsf-1*) did rescue the cold-induced phenotype (Fig. 5g–i), suggesting that intestinal *hsf-1* plays a role in *hsf-1* mutant CID. We also performed HSF-1 depletion using the auxin-inducible degron (AID) system[52]. We found that the 1 μM of auxin partially suppressed development in *hsf-1::degron* animals and 10 μM of auxin caused larval arrest at the L1 stage at 20 °C (Supplementary Fig. 6a, b). Thus, the function of HSF-1 is more strongly suppressed in *hsf-1::degron* animals treated with 1 μM auxin than in *hsf-1(sy441)* mutants. Although 1 μM-auxin treatment caused developmental arrest in *hsf-1::degron* animals at 9, 20, and 25 °C, it did not induce CID at 9 °C (Supplementary Fig. 6c–e). This result suggests that CID is specifically observed in *hsf-1(sy441)* mutants at 9 °C.

**Ageing mechanisms affect CID formation of hsf-1(sy441) mutant**
Our previous study demonstrated that the gain-of-function mutation of *hsf-90* extends lifespan in response to cold (15 °C)[5]. Moreover, we found that the *hsf-90(p673)* mutation inhibited the CID entry of *hsf-1(sy441)* mutants, but partially induced dauer formation, even at 9 °C (Fig. 6a, b, Supplementary Fig. 7a). These results suggest that the gain-of-function mutation of *hsp-90* can also restore cold acclimation in the *hsf-1* mutant.

Overexpression of *hsf-1* not only rescues developmental arrest at high temperatures but also restores lifespan[49]. A previous study reported that neuromedin U signalling also regulates lifespan[46]. Similarly, studies on dauer diapause in *C. elegans* have suggested that ageing mechanisms control entry into diapause[24,26]. Therefore, we investigated the function of known longevity mechanisms related to CID entry of the *hsf-1* mutant to examine whether CID interacts with ageing mechanisms. The expression of the spliced form of *xbp-1* in neurons activates the IRE1-XBP1 pathway in the entire body and induces *hsp-3* and *hsp-4* gene expression in *C. elegans*[53]. We found that spliced *xbp-1* expressed in neurons also prevents CID entry in the *hsf-1* mutant (Fig. 6c, e, Supplementary Fig. 7b). Overexpression of *skn-1b/c* isoforms increased lifespan at low temperatures (15 °C)[54]; therefore, we also examined *skn-1b/c* overexpression and found that *skn-1b/c* rescued CID entry in *hsf-1* mutants (Fig. 6c, e, Supplementary Fig. 7b). We further examined whether *skn-1* regulates CID using the neuronal *skn-1b* isoform[55] and determined that overexpression of *skn-1b* did not suppress CID entry (Supplementary Fig. 7c, d). This result suggests that the intestinal *skn-1c* isoform regulates *hsf-1* mutant CID[56]. Moreover, we found that overexpression of *daf-16*/FOXO transcription factor involved in insulin signalling and control of dauer formation, as well as *hlh-30*/TFEB, a key regulator of autophagy[57], also inhibited CID entry in the *hsf-1*-background mutant (Fig. 6c–e, Supplementary Fig. 7b). These

findings indicate that multiple longevity regulators play fundamental roles in the control of *hsf-1(sy441)* mutant CID.

## Genetic screening for hsf-1(sy441) mutant CID regulators

Given the interaction between CID entry and ageing mechanisms, we hypothesised that mutations preventing CID entry of the *hsf-1(sy441)* mutant may have the potential to extend lifespan; therefore, we used a mutagenesis screen to find such mutants[58]. Starting with *hsf-1(sy441)* background, we analysed ~1,000,000 EMS-mutagenised F2 genomes at 9 °C and obtained 49 non-CID strains (Fig. 7a). Next, we observed the survival ratios of these 49 non-CID strains at 15 and 20 °C and found that 16 lived longer than strains with an *hsf-1(sy441)* background mutation (Fig. 7b, Supplementary Fig. 8). Although the CID phenotype of the *hsf-1* mutant is induced by cold stimuli alone, more than half of the strains exhibited extended lifespans independent of temperature conditions. A few strains exhibited temperature-dependent long-lived phenotypes at 20 °C and 15 °C. We also confirmed the temperature-dependent phenotypes of these EMS strains and found that these strains exhibited non-CID phenotypes at 9 °C, developed normally at 20 °C, and stopped developing at 25 °C, similarly to *hsf-1* mutants (Supplementary Fig. 9a–c). These results suggest that EMS strains do not simply revert the *hsf-1(sy441)* mutation. In addition, a small number of long-lived mutants exhibited abnormalities in development and behaviour (Supplementary Table 2). As a positive control, we also tested overexpression of *skn-1b/c*, which inhibits CID entry in the *hsf-1* mutants, and found that overexpression of *skn-1b/c* slightly increased the survival ratio in the *hsf-1* background mutants (Supplementary Fig. 8).

Using the MutMap analysis outlined in Fig. 7c referring to Abe et al.[59], we sought to identify regulators of *hsf-1* mutant CID from the long-lived mutant pools and identified nonsense mutations on each quantitative trait locus (QTL) as candidates for CID regulators, including *gst-13* (EMS#28)*, mtk-1* (EMS#31)*, nsy-1* (EMS#21 and EMS#27)*, smg-1* (EMS#39)*, smg-2* (EMS#40)*, sur-2* (EMS#30 and EMS#48), and *T24E12.5/rcd-1* (EMS#49) (regulator of cold-inducible diapause) (Fig. 7d, Supplementary Fig. 10a). Reportedly, loss-of-function mutations in *smg-1* activate unfolding protein response (UPR)$^{ER}$ and increase lifespan[60,61]. *smg-1*, a mammalian homologue of the serine/threonine protein kinase SMG1, and *smg-2*, a mammalian homologue of regulator of nonsense transcripts 1, are involved in nonsense-mediated mRNA decay (NMD) and regulate longevity in several long-lived mutants[60–62]. *nsy-1* is required for intermittent fasting-related longevity in long-lived mitochondrial mutants and innate immune signalling pathways[63,64]. A previous study reported that *sur-2* is involved in memory representation in thermosensory AFD neurons[65], but its function in longevity is unclear. Furthermore, the functions of *T24E12.5/rcd-1* have not been reported. Therefore, we decided to mostly further explore the functions of the *sur-2* and *rcd-1* genes in longevity and CID entry of the *hsf-1(sy441)* mutant.

## hsf-1(sy441) mutant CID regulators control lifespan

As proof of principle, we investigated the role of *smg-1*, a reported ageing gene, in *hsf-1* mutant CID to test our hypothesis that there is an interaction between the regulatory mechanisms of ageing and CID. We used the *smg-1(cc546)* strain that has a missense mutation at different sites from *smg-1(hru112)* in the EMS#39 strain and found that the reduction-of-function mutation in *smg-1(cc546)* prevented CID entry of the *hsf-1* mutant (Supplementary Fig. 10b, 11a–c), illustrating the functional interaction between the regulatory mechanisms of *hsf-1* mutant CID and longevity. We also investigated whether *hsf-1(sy441)* mRNA is regulated by the NMD machinery by qRT-PCR. When *smg-1*-mediated NMD is involved in the quality control of *hsf-1* mRNA, *hsf-1(sy441)* mRNA is degraded by the NMD machinery, ultimately resulting in reduced *hsf-1* gene expression in the *hsf-1(sy441)* mutant. However, we found that although the *hsf-1*

mRNA was significantly increased in *hsf-1(sy441)* mutants, the *smg-1(cc546)* mutation did not affect the mRNA level of *hsf-1* in wild-type and *hsf-1(sy441)* mutants (Supplementary Fig. 11d). Furthermore, gene expression of *hsp-16.2*, a target gene of HSF-1, was not altered. This result indicates that *hsf-1(sy441)* mRNA is not regulated by the *smg-1* mediated NMD machinery and that an increase in *hsf-1(sy441)* mRNA is not sufficient to activate gene expression of HSF-1 target genes. Next, we examined the *sur-2(ku9)* strain carrying a nonsense mutation at a different site from *sur-2(hru33)* in the EMS#30 strain (Supplementary Fig. 10b). Compared to wild-type animals, *sur-2(ku9)* animals developed more slowly at cold temperatures (9 °C); nevertheless, the mutation rescued the CID phenotype in *hsf-1(sy441)* background mutants (Fig. 8a–c, Supplementary Fig. 11e). We also observed that *sur-2(ku9)* and *hsf-1; sur-2* mutants displayed a weak rod-like lethal phenotype as previously reported[66] (Supplementary Fig. 11f). Furthermore, *sur-2(ku9)* mutants lived 15% longer than their wild-type counterparts at normal temperatures (20 °C) (Fig. 8d). Next, we performed RNAi experiments targeting *rcd-1/T24E12.5*, as we were unable to obtain *rcd-1* mutants from the Caenorhabditis Genetics Centre (CGC) and the National BioResource Project (NBRP). We found that the CID phenotype of the *hsf-1* mutant was completely suppressed by RNAi treatment as well as a mutation in *rcd-1(hru150)* (Fig. 8 e-g Supplementary Fig. 11g, h). However, we did not observe long-lived phenotypes in *rcd-1(hru150)* mutants (Fig. 8 i). Therefore, we explored other mutations involved in longevity in the EMS#49 strain genome and identified a mutation in *che-11(hru153)* (Fig. 7d). A previous study reported that *che-11(e1810)* mutants harbouring a nonsense mutation are long-lived[67]. Therefore, we suggest that the EMS#49 strain has two mutations, *rcd-1(hru150)* and *che-11(hru153)*, which regulate *hsf-1* mutant CID and longevity, respectively. Although *rcd-1* was not implicated in ageing, we investigated the role of the *rcd-1* gene in detail to determine the regulatory mechanism of *hsf-1* mutant CID. We generated the *Prcd-1::rcd-1::gfp* strain and found that RCD-1::GFP was expressed in young larvae but not in adult animals (Supplementary Fig. 12). Unexpectedly, expression levels and patterns of *rcd-1* were not altered by temperature or mutation of *hsf-1*. This result suggests that the *rcd-1* gene is expressed at the time when CID is induced in the *hsf-1* mutant. We also analysed the expression of the *rcd-1* gene in wild-type, *hsf-1(sy441)* and non-CID mutants at 9°C and 20°C by qRT-PCR and found that *rcd-1* mRNA was not altered in any of the non-CID mutants and by developmental temperature (Supplementary Fig. 13a). Additionally, the gene expression of other CID regulators, including *smg-1, sur-2, nmur-1, capa-1*, and *tdc-1*, were not significantly changed during CID formation in the *hsf-1* mutant (Supplementary Fig. 13b). The *capa-1* mRNA was slightly reduced in the *hsf-1* mutant, but this is not consistent with the observation that the *capa-1* null mutation inhibited CID formation in the *hsf-1* mutant. The expression levels and patterns of *rcd-1* were not altered by temperature or mutation of *hsf-1*, and this was in agreement with the qRT-PCR results. These results suggest that the *rcd-1* gene is expressed at the time when CID is induced in the *hsf-1* mutant, but the *rcd-1* mRNA and RCD-1 protein levels were not altered during CID formation. Therefore, we believe that the function of the RCD-1 protein may be more important in the regulation of *hsf-1* mutant CID. We also examined the developmental phenotypes of the mutants of *hsf-1* mutant CID regulators in a wild-type animal background at 4°C and found that after 14 days of chilling, the eggs failed to hatch and did not recover even when returned to 20°C (Supplementary Fig. 14a, b). On the other hand, early L1 larvae of the mutants of *hsf-1* mutant CID regulators formed CID upon cold exposure at 4°C and were recovered from diapause upon warming at 20°C (Supplementary Fig. 14c, d). This result indicates that the regulators of *hsf-1* mutant CID found in this study are only involved in CID entry of *hsf-1(sy441)* mutants at 9°C, but not in wild-type CID

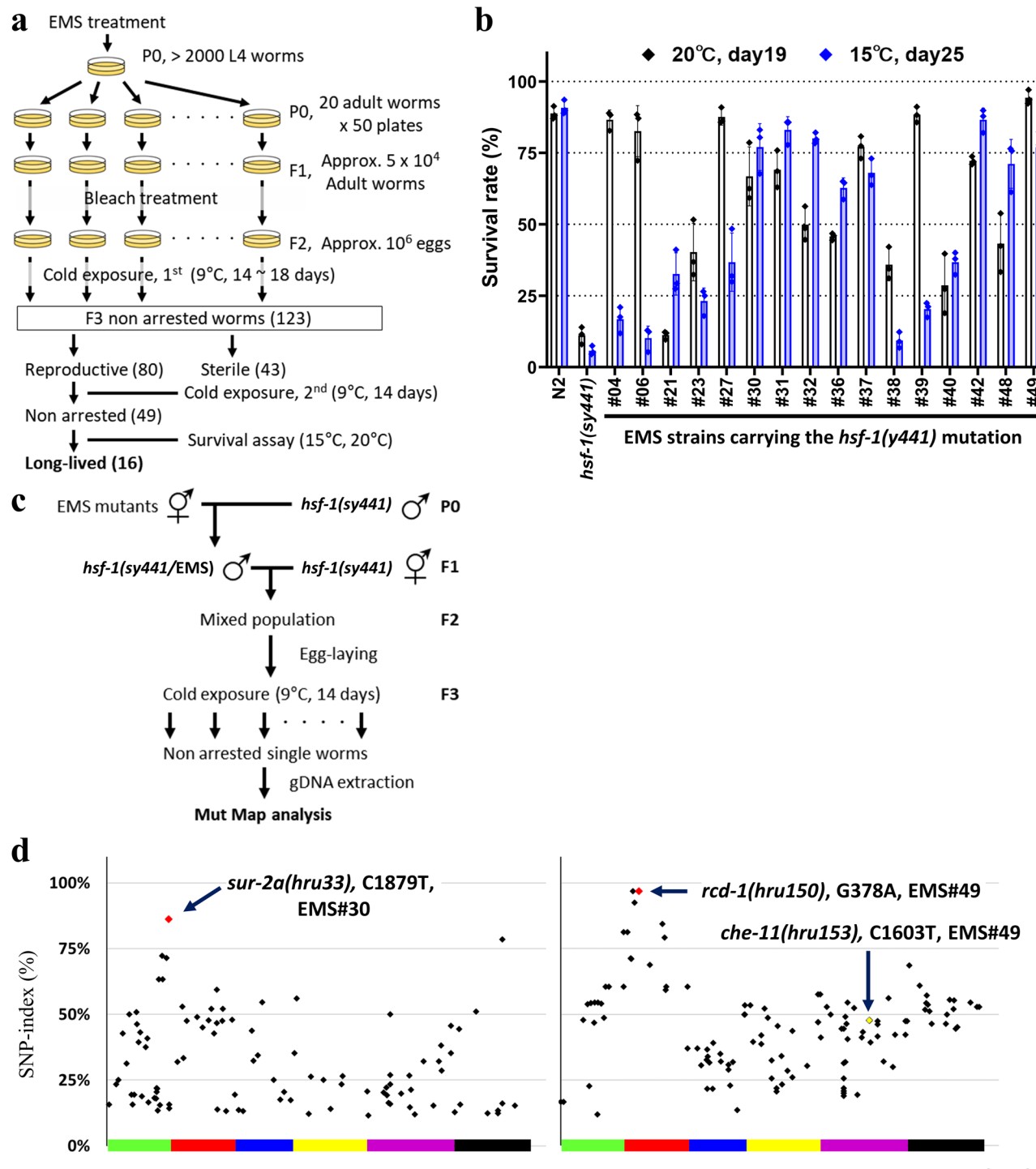

**Fig. 7 | Cold-inducible diapause can be used as a tool to screen for long-lived mutants. a** Schematic image of ethyl methanesulphonate (EMS) mutagenesis for screening of non-CID mutants. **b** The graph represents the survival ratios of long-lived mutants from the 49 non-CID mutant pool. The experiment was performed once with three technical replicates with ~50 animals. Black dots indicate the survival of each technical replicate measured on day 19 at 20 °C, and blue dots those measured on day 25 at 15 °C. The bars indicate the mean of the survival rates ± standard deviation (SD) (See also Supplementary Fig. 8). **c**, **d** Schematic image of the Mut Map analysis (**c**) and single nucleotide polymorphism (SNP)-index plots for each long-lived mutant (**d**). Red dots indicate the unique nonsense mutations in the quantitative trait loci (QTL) of each EMS mutant, whereas black dots indicate the non-synonymous SNPs. The yellow dot indicates the *che-11(hru153)* mutation. The Y-axis represents the SNP index (SNP reads/total reads). Different colours on the X-axis correspond to each chromosome. Source data are provided as a Source Data file.

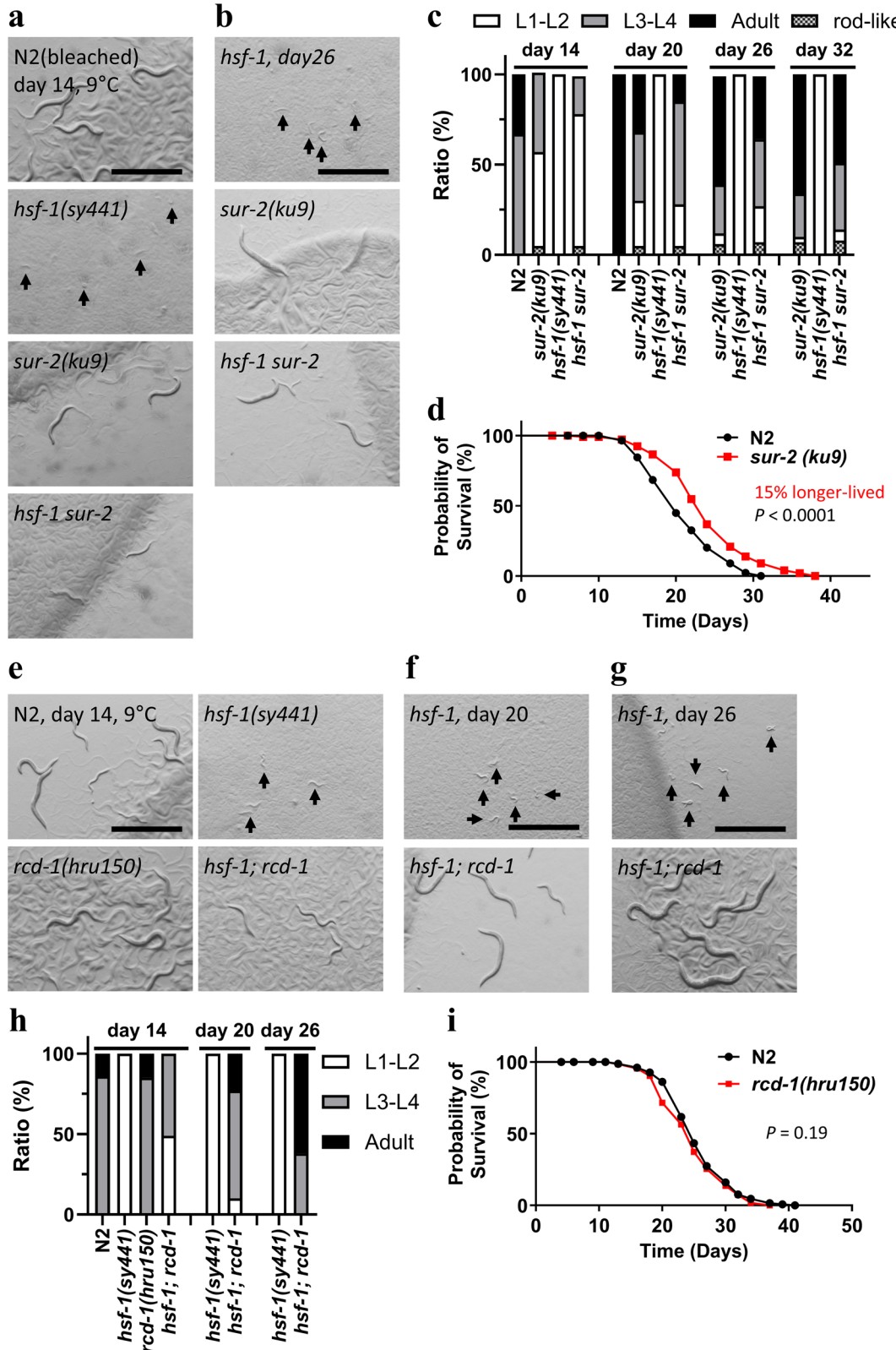

at 4°C. We also investigated the interaction of the regulatory mechanisms of dauer and *hsf-1* mutant CID. We found the *rcd-1(hru150)* mutation did not suppress dauer formation in *daf-2(e1370)* mutants (Supplementary Fig. 15a, c). We also found that a mutation in *daf-12*, a steroidal receptor and master regulator of dauer formation[24,32], did not inhibit CID formation in the *hsf-1*

mutants (Supplementary Fig. 15b, d). These results suggest that CID of *hsf-1(sy441)* mutant is regulated in parallel to dauer formation.

## Discussion

*Caenorhabditis elegans* is a powerful model organism for research on ageing, development, and thermosensation. Over the last few

**Fig. 8 | Roles of *hsf-1(sy441)* mutant CID regulator genes in longevity. a–d** The loss-of-function mutation of *sur-2*/MED23 inhibited CID induction in *hsf-1(sy441)* mutants at 9 °C and increased the lifespan of wild-type-background animals at 20 °C. **d** Black circles and red squares represent wild-type and *sur-2(ku9)* strains, respectively. Log-rank test: *P* = 0.000036 (wild-type vs *sur-2(ku9)*). **e–i** *rcd-1* mutation also inhibited CID induction in *hsf-1(sy441)* mutants at 9 °C but did not affect lifespan at 20 °C. (i) Black circles and red squares represent wild-type and *rcd-1(hru150)* strains, respectively. Log-rank test: *P* = 0.19 (wild-type vs *rcd-1(hru150)*). Ageing experiments were repeated three times with 150 animals (**d, i**). The results of

one of the three experiments are presented. For CID observations and ageing experiments, synchronised eggs were obtained using bleach treatment (**a–d**) and egg laying (**e–i**). Images were obtained on days 14 (**a, e**), 20 (**f**), and 26 (**b, g**). *n* ≥ 3 biological replicates with more than 50 animals (**a–c, e–h**). Graphs represent one of the biological replicates (**c, d, h, i**). White indicates the relative worm population of L1–L2 larvae, grey that of L3–L4 larvae, black that of adult worms, and dotted that of rod-like lethal worms (**c, h**). Scale bars, 1 mm. Arrows indicate arrested worms (**a, b, e, f, g**). Source data are provided as a Source Data file.

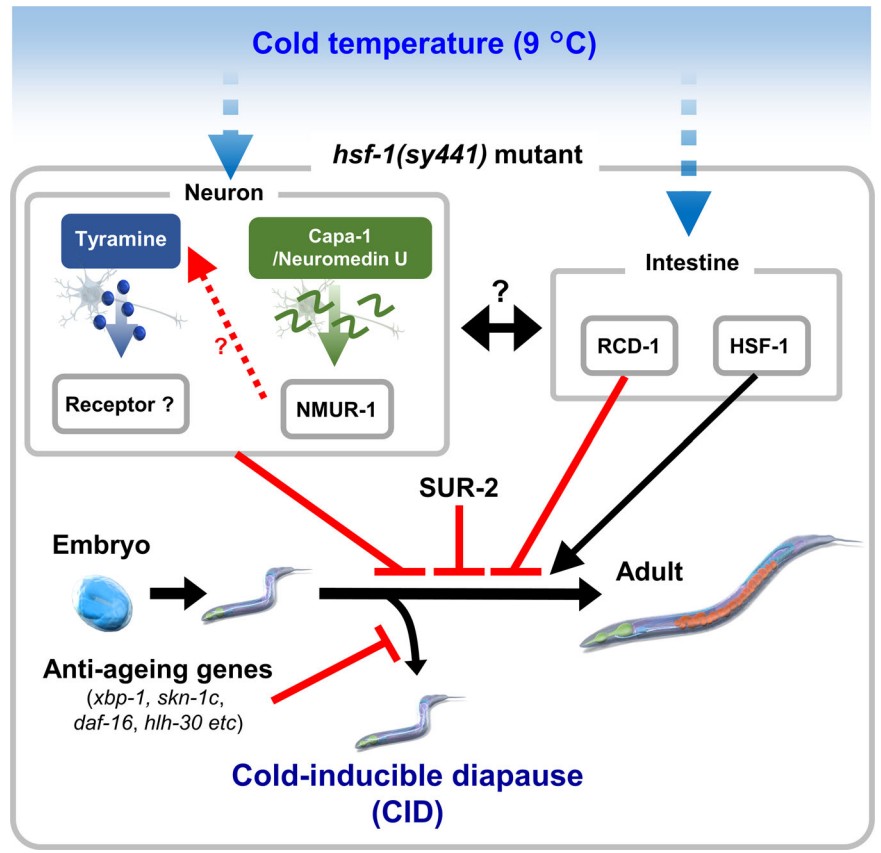

**Fig. 9 | A current working model of the regulatory mechanism of *hsf-1(sy441)* mutant CID.** In the *hsf-1(sy441)* mutant, neuromedin U and tyraminergic signalling pathways inhibit normal development at 9 °C. Intestinal *hsf-1* plays an important role in normal development at 9 °C, and *rcd-1* may trigger CID in the intestine. The

anti-ageing genes *xbp-1, skn-1c, daf-16, and hlh-30* inhibit CID entry, and *sur-2* mediates CID induction in response to cold environment. The artwork was designed by Makoto Horikawa and refined by Science Graphics. Co., Ltd. (Japan).

decades, researchers have identified neural circuits associated with temperature sensing, thermotaxis, adaptation to cold environments, and regulatory mechanisms for ageing in response to different temperatures[2,4–10,12,14,15,19,20,51] However, the interaction between short-term and long-term biological events, temperature sensing in neurons, developmental regulation, and lifespan remains poorly understood. In this study, we discovered the diapause state in *C. elegans*, termed CID, that is triggered by cold stimuli, regulated via neural circuits and under the control of several longevity mechanisms (Fig. 9). Importantly, *hsf-1* mutant CID-based analysis enables genetic screening for long-lived mutants and developmental regulators. Thus, we believe that the CID phenotype of *hsf-1(sy441)* mutant can serve as a powerful tool to investigate cold adaptation mechanisms, longevity mechanisms in response to low temperatures, and neural circuits that control development in response to the lower limit temperature of growth.

Our results demonstrate that CID is distinct from L1 arrest, dauer diapause and adult reproductive diapause. The onset of L1 arrest, dauer diapause, and adult reproductive diapause are time specific[30,31,35], but CID is induced from early larval stages L1 to L2 in *hsf-*

*1(sy441)* mutants at 9 °C and any larval stages in wild-type and *hsf-1(sy441)* mutants at 4 °C. This suggests that CID is a developmentally independent diapause phenotype. The nutritional condition is a critical factor that triggers L1 arrest, dauer diapause and ARD, whereas CID is only induced by cold environments in the presence of bacterial food. This suggests that nutritional status does not play a role in CID induction. Furthermore, we cross-checked the genetic interaction between CID and dauer formation and found that one of key regulators of *hsf-1* mutant CID, *rcd-1*, was not involved in dauer formation and *daf-12*, a master regulator of dauer formation[24,32], had no role in *hsf-1* mutant CID. These results suggest that the regulatory mechanism of CID is independent from that of other diapause states.

In addition, loss-of-function and nonsense mutants of each suppressor gene of *hsf-1* mutant CID found in this study still entered CID in a wild-type animal background at 4°C. This result suggests that these regulator genes of *hsf-1* mutant CID may be involved in altering the threshold temperature of CID entry in the *hsf-1(sy441)* mutants at 9 °C, for example, by disrupting cold sensation in the neuron or increasing cold tolerance. In contrast, 4°C is a temperature at which hatching and

development are not possible, and therefore it is difficult to analyse the regulatory mechanism. In the future, it would be useful to utilise a reporter gene of CID entry and investigate whether the regulatory mechanism of CID in the *hsf-1* mutant at 9°C is similar to or different from that of CID in wild-type animals at 4°C using this reporter.

We further investigated the function of HSF-1 in cold adaptation and unexpectedly found that the transcriptional activity of HSF-1 is not necessary for the inhibition of CID entry in the *hsf-1* mutant. We observed that *hsf-1(R145A)::gfp*, which has much lower transcriptional activity than native *hsf-1* and does not rescue the developmental arrest of the *hsf-1* mutants at 25 °C[49], rescued CID along with native *hsf-1::gfp*. We also performed HSF-1 depletion using the AID system and found that strong HSF-1 inhibition caused L1 arrest even at 20°C, as previously reported[52]. Though partial depletion of HSF-1 using *hsf-1::degron* animals with 1 μM-auxin treatment showed developmental delay at 9, 20, and 25°C, it did not, however, induce CID at 9°C. These results suggest that the CID phenotype of the *hsf-1* mutant may be specifically associated with the *hsf-1(sy441)* allele. Therefore, we propose that the transactivation domain of HSF-1, which is truncated by the *hsf-1(sy441)* mutation, plays important roles in cold adaptation and CID formation. In addition, we found that intestinal *hsf-1* is sufficient to rescue CID in *hsf-1(sy441)* mutants and that the *rcd-1* gene is also expressed in the intestine during early larval stages. Although overexpression of *skn-1b/c* inhibited CID entry in the *hsf-1* mutants, the neuronal *skn-1b* isoform did not, suggesting that the intestinal *skn-1c* isoform also regulates *hsf-1* mutant CID. These results suggest that *hsf-1, skn-1* and *rcd-1* regulate CID of the *hsf-1* mutant in the intestine as the main tissue focus.

We found that *hsp-3(ok1083)* and *hsp-4(gk514)* mutants, which are also orthologues of human ER-resident HSP70, exhibited slow growth and severe sterile phenotypes at 9 °C. Mutations in *ire-1(ok799)*, which regulates the gene expression of *hsp-3* and *hsp-4* in the IRE-1/XBP-1 pathway, also led to slow development in response to cold[53]. Furthermore, we found that the overexpression of *xbp-1s*, an active form of *xbp-1*, clearly suppressed CID formation in the *hsf-1* mutants at 9 °C. A recent study also reported that neuronal *ire-1* is required for cold acclimation at 2 °C[68]. These results suggest that the IRE-1/XBP-1 pathway plays fundamental roles not only in cold stress response at 2 °C but also in CID induction of the *hsf-1* mutant. We previously reported that a gain-of-function mutation of *hsp-90(p673)* increases lifespan at 15 °C and rescued CID in the *hsf-1* mutants at 9 °C in the present study. Several studies have reported that HSP90 forms a complex with HSF-1 and modulates the transcriptional activity of HSF-1[69,70]. These results suggest that interactions between HSP90 and HSF-1 do not only play an important role in heat stress response but also in cold adaptation and CID entry of *hsf-1(sy441)* mutant.

In this study, we demonstrated that CID of the *hsf-1(sy441)* mutant is under the control of neural pathways, similar to other diapause states[24,31]. We found that neuromedin U and tyraminergic pathways regulate entry into CID in the *hsf-1* mutant. NMUR-1, a nematode neuromedin U receptor, senses nutrient conditions and regulates longevity[46]. The *nmur-1* gene is expressed in several neurons, including the thermosensory AFD neuron and one of the tyraminergic neurons, RIC[46,71]. This suggests that the *nmur-1* gene may regulate tyramine secretion in the RIC neuron and modulate CID entry of the *hsf-1* mutant. Furthermore, CAPA-1 neuropeptide, a *C. elegans* homologue of neuromedin U, is expressed in ASG, which is a newly identified cold sensory neuron[7,45]. The DEG-1 mechanosensory receptor plays a key role in temperature sensing and is required for cold tolerance at 2 °C. These results suggest that the ASG neuron may also regulate cold acclimation by secreting the neuropeptide CAPA-1.

We observed a functional interaction between the CID phenotype of *hsf-1(sy441)* mutants and longevity genes, such as *xbp-1, skn-1, daf-16*, and *hlh-30*. Overexpression of *skn-1 b/c* is known to extend lifespan in response to cold temperatures (15 °C) and SKN-1 is involved in the regulation of unfolded protein response (UPR)[54,72]. These findings

suggest that SKN-1 may regulate cold acclimation and CID entry of the *hsf-1* mutant via the regulation of UPR chaperones, such as *hsp-3* and *hsp-4*, as well as overexpression of *xbp-1s*. We found that *daf-16* and *hlh-30* are involved in the regulation of *hsf-1* mutant CID, but the mechanism remains unclear. Although overexpression of *daf-16* inhibits *hsf-1* mutant CID, we found that a mutation of *daf-2(e1370)* did not rescue CID in the *hsf-1* mutants. A previous study reported that overexpression of *daf-16* extends lifespan as well as *daf-2(e1370)* mutants, but does not induce dauer formation at 25 °C[73]. Thus, we suggest that the functional difference between *daf-16* overexpression and the *daf-2(e1370)* mutation on development also alters its role in CID entry of *hsf-1(sy441)* mutant.

Based on our findings, we established a new mutagenesis screening approach to identify cold-response genes involved in development and longevity. By selecting for suppressors of the CID phenotype of *hsf-1(sy441)* mutant, we obtained several long-lived mutants. Compared to other genetic screening systems that use dauer and L1 arrest states, *hsf-1(sy441)* mutant CID screening has certain advantages. For example, CID can be induced in the presence of food, and therefore, the method can easily segregate non-arrested animals without special treatment, such as sodium dodecyl sulphate (SDS) selection used in Dauer selection[35]. Furthermore, worms in the CID state do not grow; therefore, the number of testable genomes is large. Unlike the regulatory mechanism underlying dauer formation, mutants exhibiting a non-CID phenotype develop normally at 9 °C, and often exhibit long-lived phenotypes at normal temperature conditions (20 °C). We successfully isolated several candidate regulators of *hsf-1* mutant CID, including *rcd-1* (regulator of cold-inducible diapause), *smg-1, smg-2, mtk-1, nsy-1, gst-13*, and *sur-2*. From these candidate genes, we demonstrated that *sur-2, smg-1*, and *rcd-1* regulate CID entry of the *hsf-1* mutant, and a mutation of *sur-2* extends lifespan. This finding indicates that mutagenesis screening using the CID phenotype of *hsf-1(sy441)* mutants can be utilised not only to explore regulators of CID but also to identify anti-ageing genes.

Reportedly, *smg-1* and *smg-2* are involved in NMD and longevity[60–62]. However, the mRNA level of *hsf-1* was significantly increased in the *hsf-1(sy441)* mutant and did not alter with a mutation of *smg-1(cc546)*. This result suggests that *smg-1* mediated NMD machinery is not involved in *hsf-1* mutant CID via quality control of *hsf-1(sy441)* mRNA. We still need to investigate how *smg-1* regulates *hsf-1* mutant CID via NMD or in parallel with NMD. The *rcd-1* gene and its paralogs, *T05A8.1* and *Y47G7B.2*, have not yet been studied, and no orthologues in higher animal species were reported on the *C. elegans* database (wormbase.org); therefore, it remains unclear how *rcd-1* is involved in CID entry of *hsf-1(sy441)* mutant.

We demonstrated that a mutation of *sur-2*, a *C. elegans* homologue of mammalian MED23[74], inhibits CID entry in *hsf-1(sy441)* mutants and extends lifespan. A recent study reported that *sur-2* is also involved in the temperature response downstream of the Raf/MAPK pathway[65]. In addition, we have identified the MAP kinase kinase kinase (MAPKKK) genes, *nsy-1* and *mtk-1*, as candidate regulators of *hsf-1* mutant CID. *nsy-1* is required for intermittent fasting-related longevity, long-lived mitochondrial mutants, and innate immune signaling pathways[63,64], but the role of *mtk-1* in ageing is unknown. Thus, we suggest that the Raf/MAPK pathway may have an important role in the regulation of CID of the *hsf-1* mutant. Reportedly, *smg-1* and *smg-2* are involved in NMD and longevity[60–62]. However, the mRNA level of *hsf-1* was significantly increased in the *hsf-1* mutant and did not alter with a mutation of *smg-1(cc546)*. This result suggests that *smg-1* mediated NMD machinery is not involved in CID of the *hsf-1* mutant via quality control of *hsf-1(sy441)* mRNA. We still need to investigate how *smg-1* regulates CID via NMD or in parallel with NMD. The *rcd-1* gene and its paralogs, *T05A8.1* and *Y47G7B.2*, have not yet been studied, and no orthologues in higher animal species were reported on the *C. elegans* database (wormbase.org); therefore, it remains unclear how *rcd-1* is involved in CID entry of *hsf-1(sy441)* mutant.

We believe that CID in the *hsf-1(sy441)* mutant can serve as a model for investigating the conserved mechanisms associated with cold-inducible dormancy such as hibernation in mammalian species and reproductive dormancy in insects. Cold stimuli activate certain neural circuits that control hibernation in vertebrate species, dormancy in invertebrate species, and CID in *C. elegans*[43,44,75]. In this study, we found that neuromedin U signalling controls CID in the *hsf-1(sy441)* mutant. Neuromedin U and hugin are reported to be involved in sleep in zebrafish[76] and *Drosophila melanogaster*[77], respectively. Thus, we suggest that part of the regulatory mechanism associated with CID is evolutionarily conserved as a quiescence mechanism in other higher animal species. Hence further investigation of the regulatory mechanism of CID should provide new insights into cold response, anti-ageing, development, the neural interaction between thermosensation and nutrient-sensing, and perhaps, mechanisms associated with hibernation.

## Methods

### Maintenance of *C. elegans* strains
All strains used in this study are listed in Supplementary Table 3. The worms were maintained using a standard worm cultivation method[78]. All mutant and transgenic strains were maintained on nematode growth medium (NGM) agar plates seeded with *E. coli* OP50 at 20 °C and maintained with sufficient bacterial food to sustain growth for at least three generations before performing each experiment. Mutants and transgenic strains were outcrossed with N2 (wild-type) animals four times before use. Mutants that were only used for pilot experiments were not outcrossed (See Supplementary Tables 1 and 3). Mutations in the strains were confirmed by PCR amplification and electrophoresis. For the genotyping of single nucleotide polymorphisms (SNPs), a combination of PCR, restriction enzyme digestion, and mismatched PCR was applied. The combinations of primer sets used, and the queried mutations are listed in Supplementary Table 4.

### Cold-inducible diapause experiment
Worms were allowed to lay eggs on NGM plates, and the hatched animals were washed with M9 buffer. Approximately 100 eggs were transferred to OP50 bacterial lawns on fresh NGM plates and cultivated at 4 °C and 9 °C. Eggs were collected using the standard bleach treatment when the experiment involved the *sur-2(ku9)* mutant. The CID phenotypes of the strains were observed, and their population was classified based on body length, and egg carrying at 14, 20, 26, and 32 days, respectively. After 14 days of incubation at 4 °C and 20, 40 and 60 days of incubation at 9 °C, recovery from CID was observed at 20 °C. Images were obtained using the binocular SZX10 (OLYMPUS) with a DP27 camera (OLYMPUS). We used ImageJ software to measure the body length of the worms.

### Morphological analysis of CID
*C. elegans* embryos and larvae were mounted on 4% agar pads. Anaesthesia was not applied in order to preserve cellular integrity and facilitate cellular identification[79]. Differential interference contrast (DIC) images were obtained using an Axio Imager M2 equipped with a Plan-Apochromat 63x/1.40 Oil DIC and Axiocam 506 mono digital camera and processed using Zen (Carl Zeiss) and Photoshop (Adobe) software.

### Auxin treatment
Worms were allowed to lay eggs on NGM plates, and the hatched animals were removed by washing the plates with M9 buffer. Approximately 100 eggs were transferred to OP50 bacterial lawns on fresh NGM plates containing 0.25, 0.5, 1.0, and 10 μM auxin (indole-3-acetic acid, Sigma) and cultivated at 9 °C, 20 °C, and 25 °C. The stock solutions of 0.25, 0.5, 1.0, and 10-mM auxin were prepared in ethanol and stored at −20 °C until use.

### Ageing experiments
Approximately 15 adult reproductive animals were transferred to OP50 plates and allowed to lay eggs for 4–6 h at 20 °C. After laying eggs, the adult animals were removed, and the synchronised population was cultivated at 20 °C until young adults were formed. Young adult animals were transferred to fresh OP50 plates containing 5-fluoro-2′-deoxyuridine (FUdR, Tokyo Chemical Industry Co, Japan) and the number of surviving worms was scored every 2–3 days. The data for sterile escaped and exploded worms were censored. The experiments were performed using blinded strains and repeated at least three times with approximately 120 animals. Statistical analyses were performed using the Kaplan-Meier method, and lifespan curves were visualised using GraphPad Prism software (GraphPad Software, Inc.).

### Mutagenesis screening
A schematic of the screening is presented in Fig. 7a. Over 2000 *hsf-1(sy441)* mutant L4 animals were collected by washing with M9 buffer in 15 mL polystyrene tubes. L4 animals were mutagenised using 0.5% ethyl methanesulphonate (EMS) in M9 buffer for 4 h with gentle shaking at room temperature ( ̃25 °C) and washed four times with M9 buffer. Mutagenised animals were cultivated overnight on OP50 plates. The following day, 20 worms from the reproductive P0 generation were transferred to 10 cm OP plates (50 plates) and maintained at 20 °C for 3 days. Adult F1 generation animals were collected in M9 buffer, and the eggs of the F2 generation were obtained using the standard bleach method[78]. The eggs were seeded to obtain approximately 5000 eggs/plate on 10 cm OP50 plates, and CID was induced at 9 °C. Plates were observed on days 14, 16, and 18, and animals larger than L4 larvae were singled into fresh OP plates. These non-CID strains were grown at 20 °C and a secondary CID screen was performed on their progenies.

The survival assay was performed using non-CID strains from the 2nd screen at 15 °C and 20 °C. Non-CID strains were prepared using a standard protocol for ageing experiments, as described in this manuscript. The survival ratio was scored on day 19 at 20 °C, and day 25 at 15 °C. The assay was performed once with three technical replicates.

### MutMap analysis
Each non-CID strain was outcrossed twice with *hsf-1(sy441)*-background mutants. Eggs from the F3 generation were obtained from a mixed population of the F2 generation and CID was induced at 9 °C. Non-CID animals were segregated and maintained on fresh OP plates, and their progenies were re-tested for CID. Descendants exhibiting the same frequency of the non-CID phenotype as the original mutagenised strains were collected, the population was expanded, and genomic DNA was extracted using the ISOSPIN tissue DNA kit (NIPPON GENE, Japan). For each non-CID mutant line, equal amounts of genomic DNA from outcrossed descendants were mixed and whole genome sequencing was performed by Novogene using the Illumina Novaseq™ 6000. Data analysis was also performed by Novegene. The reference sequence WGS data of *hsf-1(sy441)* was subtracted from the WGS data of each mutagenised strain, and SNP-index plots were generated based on density maps of single nucleotide variants. For each QTL, nonsense mutations were searched and four candidate CID regulators from individual mutagenised strains were identified. Mutagenised strains that did not carry nonsense mutations in the QTL were not analysed further in the study.

### Heat shock assay
The worms were prepared using the same protocol described above for the CID experiments. Approximately 100 eggs were transferred to OP50 bacterial lawns on fresh NGM plates and cultivated at 20 °C and 25 °C. The dauer formation of the strains was observed at 60, 84 (25 °C), and 72 h (20 °C), respectively.

## Statistics analysis

No statistical method was used to predetermine the sample size. No data were excluded from the analyses. For the ageing experiment, adult animals were randomly transferred from the population to the test plates and the worm strains were blinded. In other experiments, embryos were randomly transferred from the population to the test plates, but the worm strains were not blinded. Kaplan-Meier statistics was used to analyse the ageing experiment. *P*-value < 0.05 is considered significant.

## Reporting summary

Further information on research design is available in the Nature Portfolio Reporting Summary linked to this article.

## Data availability

Identifiers of nematode strains and a bacterial strain that was used in this study are listed in Supplementary Table 3. The WGS data of the EMS strains generated in this study have been deposited in the DDBJ BioProject database under accession code PRJDB16890. Source data are provided in this paper.

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

## Acknowledgements

We thank the current and former members of the Mizunuma and Antebi laboratories. The authors thank Dr. Keith Blackwell (Joslin Diabetes Centre) and Dr. Ikue Mori (Nagoya University) for providing the strains used in this study. Strains were also provided by the Caenorhabditis Genetics Centre (CGC), which is funded by the Office of Research Infrastructure Programmes (P40 OD010440), and by the National BioResource Project (NBRP), which is funded by the Japanese government. The authors thank Dr. Ryuichi Hirota and Dr. Akio Kuroda for assisting with the microscopic analysis. The authors also thank Dr. Yidong Shen, Dr. Shuhei Nakamura, Dr. Seung-Jae Lee, and Dr. Takashi Ito for their valuable advice on this project. Library preparation and WGS were performed by Novogene using the Illumina Novaseq™ 6000. The authors thank Novogene for

providing technical support. This work was supported by JSPS KAKENHI Grant Numbers JP22K06596 (M.H), JP22H02260 (M.M), and AMED Grant Numbers JP22gm6110029 (M.M), and 21gm6110017h0004 (M. F.). This work was also funded by the Naito Foundation (M.H), Koyanagi Foundation (M.H), Takaki Shunsuke Foundation For Science and Technology of Bread (M.M), and the Shiraishi Foundation of Science Development (M.M). The work was conducted with the facilities in the Natural Science Centre for Basic Research and Development (N-BARD) at Hiroshima University (NBARD-00157). We also thank Hiroshima University for their financial support in publishing this paper.

## Author contributions

M.H. designed and performed the experiments, analysed the data, and generated the strains used in this study. M.F. performed the morphological analysis of the CID. M.H. wrote the manuscript, along with the input of several suggestions from M.F., A.A., and M.M. supervised the project. All authors contributed to the revision of the manuscript.

## Competing interests

The authors declare no competing interests.
