## [Peer Review File · Nature Communications]

Regulatory mechanism of cold-inducible diapause in
Caenorhabditis elegansEditorial Note: Parts of this Peer Review File have been redacted as indicated to maintain the confidentiality of unpublished data.

REVIEWER COMMENTS

Reviewer #1 (Remarks to the Author):

This manuscript presents the genetic characterization of a *C. elegans* larval arrest under cold temperatures and focus on the role of *hsf-1* as a gene that regulates a larval arrest termed cold induced diapause (CID). These studies include the identification of suppressors of *hsf-1* for this larval arrest phenotype, some of which are also involved in ageing. From this list of suppressors, the authors highlight those involved in neural signalling as well as a new gene (*rcd-1*).

Major issues

1. There are important aspects of CID that are unclear and need to be clarified so that the *hsf-1* phenotype is interpretable.

(a) Is CID a normal physiological response regulated by *hsf-1* or a pathological condition seen only in *hsf-1* mutants? Do wild-type animals undergo CID at 4°C? The authors mention that wild-type animals arrest at 4°C whereas *hsf-1* mutants arrest at 9°C. Is it because *hsf-1* makes the animals more cold-sensitive? These questions should be addressed by morphological comparisons of wild-type at 4°C and *hsf-1* mutants at 9°C, and by testing if the genetic manipulations that allow *hsf-1* mutants to bypass CID at 9°C also allows *hsf-1*(+) animals to grow up at 4°C.

Determining if wild-type animals undergo CID at 4°C is critical for interpreting the *hsf-1* phenotype. If *hsf-1* mutants are adopting a normal form of arrest but at a higher temperature, it would indicate that *hsf-1* is modulating the temperature-sensitivity of a normal physiological process, like *daf-2*'s role in L1 arrest or dauer formation. Alternatively, if CID is not a process adopted by wild-type animals regardless of temperature, it would suggest that CID is a *hsf-1*-specific pathology.

(b) Addressing (a) above is important also because it can shed light on the basis of CID. Previous work (Chan et al., MBoC 2010 <https://doi.org/10.1091/mbc.e09-07-0614>) has shown that *C. elegans* embryos arrest at 4°C due to chromosome segregation defects and this phenotype can be suppressed by anoxia. If wild-type animals undergo CID at 4°C and *hsf-1* is shifting the temperature

sensitivity, then it may suggest potential roles for suppressors of hsf-1 identified throughout this manuscript in chromosome segregation, other cell cycle processes, or anoxia.

2. The relationships among hsf-1 and its suppressors should be clarified. The results are mostly a list of genes that suppress hsf-1 without much definitive biological bases for their roles or how these suppressors interact functionally and/or regulate one another. The model proposed (e.g., in Figure 4I) remains very speculative, and many alternative models are compatible with the results.

From first principles, a suppressor could act genetically downstream or in parallel of hsf-1. Figure 4I indicates that tdc-1 and/or capa-1 act in parallel to hsf-1. However, the results do not rule out alternative models where tdc-1 and/or capa-1 act downstream of hsf-1, and where hsf-1 acts from the intestine to affect neural functions such as tyramine and/or neuromedin U signalling.

[Redacted]

These relationships among various suppressors of hsf-1 and with hsf-1 need to be clarified by analysis of genetic and regulatory interactions between these suppressors. These experiments are required to show whether one suppressor regulates the expression and/or activity of another suppressor to modulate CID. Only with such further analysis can it be possible to progress from a list of genes to a more coherent picture of how these genes work together to regulate CID.

Minor Issues

3. On page 3, the text indicates: “By narrowing the lowest threshold temperature, wild-type worms could develop and reproduce at 9°C but ceased growth at 4°C.” However, there is no data or methods supporting this. More details should be provided including the number of animals and trials, and temperatures tested between 4°C and 9°C.

4. The methods section does not indicate what temperatures the parent worms were cultivated at. Since prior cultivation temperatures can affect survival and growth at low temperatures in some circumstances (Ohta et al., Nature Comms 2014 <https://www.nature.com/articles/ncomms5412>), these experimental details should be reported.

5. Gain-of-function experiments (overexpression) is used to implicate several anti-aging genes in CID. However, such experiments do not address whether these genes have an endogenous role in

the process. Furthermore, although daf-16 overexpression can overcome CID, a daf-2 mutation that is known to endogenously increase DAF-16 activity do not. Such results raise the possibility that the suppression of hsf-1 induced CID by daf-16 is not physiological. This discrepancy should be addressed by stating the caveats of these experiments in the text.

6. On page 6 “These results indicate that neuron-to-intestine communication regulates CID entry...” As described in point 2 above, these results do not necessarily indicate neuron-to-intestine communication. Since there is no data showing that these neural pathways regulate hsf-1, the results could equally reflect intestine-to-neuron signalling. This conclusion should be left out or be addressed with experiments that assess hsf-1 gene or protein activity.

7. [Redacted]

8. Strain identifiers and allele identifiers must be provided for all strains and alleles based on standard *C. elegans* nomenclature. Many strains in Supplementary Table 3 lack a standard strain identifier. The allele designations rcd-1(G394A), che-11(C1603T), and sri-47(G39A) also do not follow standard *C. elegans* nomenclature.

9. The method section indicates that all strains used were outcrossed four times (page 15), but many of the strains listed are original strains from knockout consortia which have not been outcrossed. For example, the manuscript indicates that ire-1(ok799) was tested for growth at 9°C, but the only ire-1 strain listed in RB925 which is not outcrossed. This issue affects many strains listed in Supplementary Table 3 and must be clarified so that it is clear as to the strain that was used for the experiments and that only outcrossed strains were used.

Reviewer #2 (Remarks to the Author):

In this study Makoto Horikawa et al identified a new diapause state in *C. elegans*. This diapause state (CID in short) is induced by cold, especially in the presence of mutated HSF-1 (the sy441 allele), unaffected by the presence of food and is reversible upon temperature shift. CID of hsf-1 mutants requires neuropeptide (CAPA-1) and neurotransmitter (Tyramine). CID of hsf-1 mutants can be suppressed by intestinal expression of HSF-1 as well as by OE of a variety of stress/aging-related transcription factors including daf-16, xbp-1s, hlh-30 and skn-1. Finally, the authors perform an EMS suppressor screen, and isolate nearly 50 hsf-1 mutant strains that fail to undergo CID. Many of

these strains were also long-lived relative to the hsf-1 strain, suggesting a possible correlation between CID and longevity regulatory mechanisms.

The identification of a new diapause state in *C. elegans* is interesting and innovative, as is the dissection of the under-studied stress response to cold. I appreciate that the study addresses many CID-related biological questions, however this wide-scope analysis is not accompanied by sufficient in-depth analysis, that really nails the underlying mechanism. Another, major issue is that most of the analysis of CID is done in hsf-1 mutants rather than in WT animals. If the phenomenon is specific to the mutant background, and does not relate to the WT strain – this significantly reduces its significance. I would expect CID to occur in WT animals as well (perhaps at lower temperature?). I would also expect to see some inhibition of HSF-1 pathway upon cold exposure (this could make the hsf-1 data relevant to WT animals).

To summarize, the work is innovative and of high potential, however more in-depth analysis is required.

Major comments:

1) The introduction and discussion sections of the paper are each single paragraphs. This is not sufficient to layout sufficient background information for the readers, and not enough to present the breakthroughs and importance of the work. These sections need to be written and expended. At the moment they practically do not exist. I would expect to see at minimum comparisons to other diapause states in *C. elegans*, comparison to other cold stress responses in *C. elegans*, and previous studies implicating HSF-1/IRE-1 in cold stress response. I would expect to get background information about the main player here- HSF-1.

2) The identification of a new diapause state is important. I expect to see a more thorough physiological comparison to other diapause states of *C. elegans* at the experiment level.

3) The CID phenomenon must be connected to WT physiology (rather than to the physiology of the hsf-1 mutant). If the phenomenon is specific to a small set of mutants, it becomes esoteric. Are there more extreme cold conditions that induce CID in WT animals? In the 1st paragraph of the results section it is mentioned that WT worms ceased growth at 4 degrees. Is this CID? If so, CID should be characterized on WT animals rather than than on hsf-1 mutants.

4) HSF-1 regulation by cold should be investigated. Is HSF-1 activity downregulated in any way under these cold stress conditions? Is it spread in the nucleus or generates foci? Are cytosolic chaperones transcribed?

5) The HSF-1(sy441) mutation is an early stop codon which terminates the protein prior to its transactivating domain. Nevertheless, the resulting truncated HSF-1 is still functional, and may represent a gain of function state (PMID: 25324391). Can CID be similarly achieved with hsf-1 RNAi treatment?

6) The authors mention several times that CID is food independent. While food is obviously present – are you sure that enough food enters the animals?

7) The lack of CID suppression by daf-2 mutation, while effectively suppressing the phenotype by DAF-16 OE is contradictory and surprising. Any speculations to this contradiction?

8) For the Putative CID suppressive mutations identified in the EMS screen – I would expect to confirm the relevance of all of the candidate mutations in CID and/or longevity using independent mutants (generated by CRISPR, or available from other sources), and to be outcrossed several times to remove background mutations. Without showing that the CID suppression and longevity phenotype stem from the same gene – it is impossible to argue correlation/enrichment.

9) [Redacted]

10) Smg-1 and smg-2 are part of the NMD pathway. Hsf-1(sy441) carries a premature stop codon and thus is a potential substrate of the NMD pathway. One must verify that the suppression by smg-1/2 is not simply due to stabilization of the hsf-1(sy441) transcript.

11) The authors think rcd-1 might be a master regulator of CID. If so, shouldn't it be characterized more? Does it act upstream/downstream/in parallel to HSF-1?

Minor comments:

1) All aging-related genes tested may also be defined as stress-related genes. One should be more careful when associating them with one of these linked phenotypes.

2) Page 6 lines 180-181 - the names of the tissue specific promoters were switched.

3) Reference 33 seems to be the wrong reference

4) Page 6 line 215 – the skn-1 OE result seems out of place

5) It is unclear how specific neuropeptides/neurotransmitters were selected for analysis.

Reviewer #3 (Remarks to the Author):

In this study, Horikawa et al, demonstrate that cold-inducible diapause (CID), which is induced by low temperature at 9 degrees celsius is regulated by HSF-1. Interestingly, a normally short-lived mutant of HSF-1, hsf-1(sy441) survives longer in this state of diapause. The authors used extensive epistasis analysis along with a forward genetic screen to identify regulators of this response. They identify several, including transcription factors required for proteostasis, as well as a neuron-to-intestine signaling pathway that via tyraminerbic signaling seems to be important for this response. While this paper is very interesting and intriguing, it also appears a bit preliminary. My major concern is that the take home message is muddled by too many factors and parallel signaling pathways involved in this response. After a lot of epistasis analysis it is still not clear how this is regulated or if there is a core pathway that regulates this from the neurons to the intestine. A lot of

conclusions they are making are not really supported by the evidence (please see major questions below).

What I am also missing is a more extended Discussion, where I would have expected at least some mention of the implications for this. The authors mention the requirement for neuromedin U signaling for cold acclimation and thermogenesis in vertebrates, without further expanding on its importance. Perhaps the intestine is an important organ regulating thermogenesis as a fat-storage organ in *C. elegans*.

A lot of open questions remain:

1) The authors say the heat shock response is required for this. But it seems more like the UPR (Ire-1/xbp-1 pathway) is required and ER resident Hsp70 (BiP) is induced rather than HSF-1 regulated hsp-70 (C12C8.1). More analysis is required to clarify this.

2) The *daf-2* mutant does not rescue the CID phenotype of *hsf-1* mutant and *daf-2* mutants also do not enter CID. The authors suggest that *daf-16* overexpression rescues this. If not regulated via the DAF-2 signaling pathway, how is *daf-16* then a regulator of CID?

3) **[Redacted]**

4) If *rcd-1* is a master regulator of CID, an epistasis analysis of other genes highlighted in this report is appropriate. For example, with the mentioned heat shock proteins or down-stream transcription factors.

5) Where is *nmur-1*, *rcd-1* and *sur-1* expressed? This would be relevant to highlight importance of the neuron-to-intestine signaling response identified here.

6) **[Redacted]**

Minor:

Page 3, line 101: hsp-70 is the heat-inducible Hsp70 isoform, also known as C12C8.1 in *C. elegans*. The nomenclature “hsp-70” used here seems wrong and should be changed to Hsp70 if talking

generally about Hsp70 and its orthologues. The Hsp70 orthologue regulated by the IRE-1/XBP-1 pathway here is the ER-resident Hsp70, also known as BiP or hsp-4 in *C. elegans*.

Page 6, line 181 and 182: pan-neuronal expression of hsf-1 should be *unc-14p::hsf-1* and intestinal expression of hsf-1 should be *ges-1p::hsf-1*. This has been switched here.

Reviewer #1 (Remarks to the Author):

This manuscript presents the genetic characterization of a *C. elegans* larval arrest under cold temperatures and focus on the role of *hsf-1* as a gene that regulates a larval arrest termed cold induced diapause (CID). These studies include the identification of suppressors of *hsf-1* for this larval arrest phenotype, some of which are also involved in ageing. From this list of suppressors, the authors highlight those involved in neural signalling as well as a new gene (*rcd-1*).

➤ We thank you for your many valuable comments.

Major issues

1. There are important aspects of CID that are unclear and need to be clarified so that the *hsf-1* phenotype is interpretable.

(a) Is CID a normal physiological response regulated by *hsf-1* or a pathological condition seen only in *hsf-1* mutants? Do wild-type animals undergo CID at 4°C? The authors mention that wild-type animals arrest at 4°C whereas *hsf-1* mutants arrest at 9°C. Is it because *hsf-1* makes the animals more cold-sensitive? These questions should be addressed by morphological comparisons of wild-type at 4°C and *hsf-1* mutants at 9°C, and by testing if the genetic manipulations that allow *hsf-1* mutants to bypass CID at 9°C also allows *hsf-1*(+) animals to grow up at 4°C.

Determining if wild-type animals undergo CID at 4°C is critical for interpreting the *hsf-1* phenotype. If *hsf-1* mutants are adopting a normal form of arrest but at a higher temperature, it would indicate that *hsf-1* is modulating the temperature-sensitivity of a normal physiological process, like *daf-2*'s role in L1 arrest or dauer formation. Alternatively, if CID is not a process adopted by wild-type animals regardless of temperature, it would suggest that CID is a *hsf-1*-specific pathology.

➤ We thank you for your valuable comments and conducted a phenotypic analysis at 4°C. As shown in Fig 2a, wild-type and *hsf-1*(*sy441*) mutant embryos were completely killed by cold exposure at 4°C for 2 weeks. However, we discovered that both wild-type and *hsf-1*(*sy441*) mutants are able to enter CID during 4°C-cold exposure in the early L1 stage, the timing at which *hsf-1*(*sy441*) mutants form CID (Fig 2 b). Interestingly, *hsf-1*(*sy441*) mutants recovered slowly from CID induced at 4°C and the recovered animals exhibited several morphological abnormalities; sterility, egg-laying defects and protruding vulva (Fig 2 b,c). These differences suggest that *hsf-1*(*sy441*) mutants are more sensitive to cold than the wild-type animals and

enter CID at a higher temperature (9°C). Furthermore, we found that wild-type and *hsf-1(sy441)* mutants can enter CID at all developmental stages (L1 to L4) (Fig 3 c,d). We examined all mutations that inhibit CID formation in *hsf-1(sy441)* mutants, such as *rcd-1(hru150)* at 4°C, and found that these mutants, namely *rcd-1(hru150)*, *nmur-1(ok1387)*, *capa-1(ok3065)*, *tdc-1(ok914)*, and *smg-1(cc546)*, could enter CID and recover from CID after a temperature shift to 20°C, similar to wild-type animals (data not shown). These results suggest that these genes may regulate CID induction by regulating the temperature sensitivity of *hsf-1(sy441)* mutants.

(b) Addressing (a) above is important also because it can shed light on the basis of CID. Previous work (Chan et al., MBoC 2010 <https://doi.org/10.1091/mbc.e09-07-0614>) has shown that *C. elegans* embryos arrest at 4°C due to chromosome segregation defects and this phenotype can be suppressed by anoxia. If wild-type animals undergo CID at 4°C and *hsf-1* is shifting the temperature sensitivity, then it may suggest potential roles for suppressors of *hsf-1* identified throughout this manuscript in chromosome segregation, other cell cycle processes, or anoxia.

➤ Thank you for your valuable comment. As specified in the response above, both wild-type and *hsf-1(sy441)* mutant embryos were unhatched and killed at 4°C, but hatched larvae could enter CID. Therefore, we suggest that the regulatory mechanism of CID may differ to that associated with embryonic arrest reported in a previous work (Chan et al., MBoC 2010).

2. The relationships among *hsf-1* and its suppressors should be clarified. The results are mostly a list of genes that suppress *hsf-1* without much definitive biological bases for their roles or how these suppressors interact functionally and/or regulate one another. The model proposed (e.g., in Figure 4I) remains very speculative, and many alternative models are compatible with the results. From first principles, a suppressor could act genetically downstream or in parallel of *hsf-1*. Figure 4I indicates that *tdc-1* and/or *capa-1* act in parallel to *hsf-1*. However, the results do not rule out alternative models where *tdc-1* and/or *capa-1* act downstream of *hsf-1*, and where *hsf-1* acts from the intestine to affect neural functions such as tyramine and/or neuromedin U signalling.

[Redacted]

These relationships among various suppressors of *hsf-1* and with *hsf-1* need to be clarified by analysis of genetic and regulatory interactions between these suppressors. These experiments are required to show whether one suppressor regulates the expression and/or activity of another suppressor to modulate CID. Only with such further analysis can it be possible to progress from

a list of genes to a more coherent picture of how these genes work together to regulate CID.

- Thank you for your valuable comment. [Redacted] To investigate the genetic interaction between *hsf-1* genes and putative CID regulator genes, we focused on *rcd-1*, which requires CID formation in *hsf-1(sy441)* mutants and generated *Prcd-1::rcd-1::GFP* transgenic strains. However, the gene expressions and patterns of *rcd-1* were not altered by environmental temperatures and mutation of *hsf-1(sy441)* (Extended data Fig 11). Therefore, we believe that RCD-1 regulates CID at the protein level rather than at the gene expression level. We also crossed mutants of putative CID regulator genes with *Phsp-4::gfp*, one of the UPR chaperones, and *Phsp16.2::gfp*, a major target gene of HSF-1, transgenic strains; however, did not observe any changes in gene expression in these strains (Data not shown). Therefore, we are now exploring other marker genes that reflect CID induction. With our current data, we cannot infer a hierarchical interaction between the neural network and the intestine in CID formation. Therefore, we withdrew the previous working model as per your suggestion and have presented a more simplified model (Fig 9).

Minor Issues

3. On page 3, the text indicates: “By narrowing the lowest threshold temperature, wild-type worms could develop and reproduce at 9°C but ceased growth at 4°C.” However, there is no data or methods supporting this. More details should be provided including the number of animals and trials, and temperatures tested between 4°C and 9°C.

- Thank you for your valuable comment. We only tested 4°C and 9°C for the analyses. We have presented the results in Fig 2a and have described the experimental conditions in the legend, on page 25, lines 883-892.

4. The methods section does not indicate what temperatures the parent worms were cultivated at. Since prior cultivation temperatures can affect survival and growth at low temperatures in some circumstances (Ohta et al., Nature Comms 2014 <https://www.nature.com/articles/ncomms5412>), these experimental details should be reported.

- Thank you for your valuable comment. We investigated the effect of maternal growth temperature on CID induction and found that it did not alter the development of wild-type and *hsf-1(sy441)* mutants (Extended data Fig 2e). Wild-type animals did not enter CID when pre-incubated at 15, 20 and 25 °C. *hsf-1(sy441)* mutants entered CID when pre-incubated at

15 and 20 °C. We did not analyze *hsf-1(sy441)* mutants at 25 °C because they exhibited a developmental arrest phenotype (as Fig 5 d).

5. Gain-of-function experiments (overexpression) is used to implicate several anti-aging genes in CID. However, such experiments do not address whether these genes have an endogenous role in the process. Furthermore, although *daf-16* overexpression can overcome CID, a *daf-2* mutation that is known to endogenously increase DAF-16 activity do not. Such results raise the possibility that the suppression of *hsf-1* induced CID by *daf-16* is not physiological. This discrepancy should be addressed by stating the caveats of these experiments in the text.

➤ We agree with your comment. We have explained the discrepancy between overexpression of *daf-16* and the *daf-2* mutation on page 14, lines 472-476. “A previous study reported that overexpression of *daf-16* extends lifespan as well as *daf-2(e1370)* mutants, but does not induce dauer formation at 25 °C⁷¹. Thus, we suggest that the functional difference between *daf-16* overexpression and the *daf-2(e1370)* mutation on development also alters its role in CID entry.”

6. On page 6 “These results indicate that neuron-to-intestine communication regulates CID entry...” As described in point 2 above, these results do not necessarily indicate neuron-to-intestine communication. Since there is no data showing that these neural pathways regulate *hsf-1*, the results could equally reflect intestine-to-neuron signalling. This conclusion should be left out or be addressed with experiments that assess *hsf-1* gene or protein activity.

➤ We agree with your comment that we have not demonstrated “neuron-to-intestine communication” clearly in this work, and therefore, decided to remove the sentence from the manuscript.

7. [Redacted]

➤ [Redacted]

8. Strain identifiers and allele identifiers must be provided for all strains and alleles based on standard *C. elegans* nomenclature. Many strains in Supplementary Table 3 lack a standard strain identifier. The allele designations *rcd-1(G394A)*, *che-11(C1603T)*, and *sri-47(G39A)* also do not follow standard *C. elegans* nomenclature.

➤ Thank you for your valuable comment. We registered our laboratory code “MZ” and allele code “*hru*” via wormbase and modified the text, figures and Supplementary Table 3 with the standard *C. elegans* nomenclature.

9. The method section indicates that all strains used were outcrossed four times (page 15), but many of the strains listed are original strains from knockout consortia which have not been outcrossed. For example, the manuscript indicates that *ire-1(ok799)* was tested for growth at 9°C, but the only *ire-1* strain listed in RB925 which is not outcrossed. This issue affects many strains listed in Supplementary Table 3 and must be clarified so that it is clear as to the strain that was used for the experiments and that only outcrossed strains were used.

➤ Thank you for your valuable comment. We used different strain names for outcrossed strains and have updated Supplementary Table 3. We used some strains that did not exhibit abnormalities at 9 °C without outcrossing. We have modified Supplementary Table 3 and the methods on page 15, lines 530-532.

Reviewer #2 (Remarks to the Author):

In this study Makoto Horikawa et al identified a new diapause state in *C. elegans*. This diapause state (CID in short) is induced by cold, especially in the presence of mutated *HSF-1* (the sy441 allele), unaffected by the presence of food and is reversible upon temperature shift. CID of *hsf-1* mutants requires neuropeptide (CAPA-1) and neurotransmitter (Tyramine). CID of *hsf-1* mutants can be suppressed by intestinal expression of *HSF-1* as well as by OE of a variety of stress/aging-related transcription factors including *daf-16*, *xbp-1s*, *hlh-30* and *skn-1*. Finally, the authors perform an EMS suppressor screen, and isolate nearly 50 *hsf-1* mutant strains that fail to undergo CID. Many of these strains were also long-lived relative to the *hsf-1* strain, suggesting a possible correlation between CID and longevity regulatory mechanisms.

The identification of a new diapause state in *C. elegans* is interesting and innovative, as is the dissection of the under-studied stress response to cold. I appreciate that the study addresses many CID-related biological questions, however this wide-scope analysis is not accompanied by sufficient in-depth analysis, that really nails the underlying mechanism. Another, major issue is that most of the analysis of CID is done in *hsf-1* mutants rather than in WT animals. If the phenomenon is specific to the mutant background, and does not relate to the WT strain ? this significantly reduces its significance. I would expect CID to occur in WT animals as well (perhaps at lower temperature?). I would also expect to see some inhibition of *HSF-1* pathway upon cold exposure (this could make the *hsf-1* data relevant to WT animals).

To summarize, the work is innovative and of high potential, however more in-depth analysis is required.

➤ We thank you for your many valuable comments.

Major comments:

1) The introduction and discussion sections of the paper are each single paragraphs. This is not sufficient to layout sufficient background information for the readers, and not enough to present the breakthroughs and importance of the work. These sections need to be written and expanded. At the moment they practically do not exist. I would expect to see at minimum comparisons to other diapause states in *C. elegans*, comparison to other cold stress responses in *C. elegans*, and previous studies implicating *HSF-1*/IRE-1 in cold stress response. I would expect to get background information about the main player here- *HSF-1*.

➤ Thank you for your valuable comment. We have expanded the Introduction and Discussion sections of the revised manuscript.

2) The identification of a new diapause state is important. I expect to see a more thorough physiological comparison to other diapause states of *C. elegans* at the experiment level.

➤ Thank you for your valuable comment. We compared CID with L1 arrest in Fig 1 e. Starvation is an environmental trigger of L1 arrest; however *hsf-1(sy441)* mutants have shorter lifespans under starvation conditions. CID is induced in *hsf-1(sy441)* mutants even in the presence of food. These results suggest that CID is distinct from L1 arrest. We also analyzed *rcd-1*, a regulator of CID, in dauer-constitutive *daf-2(e1370)* mutants and found that the *rcd-1* mutation did not inhibit dauer formation (Extended data Fig 12). We also analyzed *daf-12*, a master regulator of dauer formation, in *hsf-1(sy441)* mutants and found that *hsf-1; daf-12* mutants entered CID similarly to *hsf-1(sy441)* mutants. These results suggest that the regulatory mechanisms of CID and dauer formation are in parallel. Furthermore, we found that CID was induced at all developmental stages (from L1 to L4) at 4 °C (Figure 3 c). These results suggest that CID can be induced at multiple developmental stages; therefore, differs from other diapause states.

3) The CID phenomenon must be connected to WT physiology (rather than to the physiology of the *hsf-1* mutant). If the phenomenon is specific to a small set of mutants, it becomes esoteric. Are there more extreme cold conditions that induce CID in WT animals? In the 1st paragraph of

the results section it is mentioned that WT worms ceased growth at 4 degrees. Is this CID? If so, CID should be characterized on WT animals rather than than on *hsf-1* mutants.

- Thank you for your valuable comment. As shown in Figures 2, 3 and extended Figure 3, wild-type and *hsf-1(sy441)* mutant embryos failed to hatch and were killed at 4°C. However, wild-type and *hsf-1(sy441)* mutant larvae that hatch at 20°C entered CID after being exposed to 4°C.

4) *HSF-1* regulation by cold should be investigated. Is *HSF-1* activity downregulated in any way under these cold stress conditions? Is it spread in the nucleus or generates foci? Are cytosolic chaperones transcribed?

- Thank you for your valuable comment. We performed a more detailed functional analysis of HSF-1. As a previous study reported (Elizabeth A. Morton, AgingCell. 2013), HSF-1 is inactive at 20°C. This suggests that HSF-1 is not further downregulated by cold temperatures. This previous study also reported that trimerization of HSF-1 is required for heat-stress resistance and we demonstrated that native *hsf-1* alleviates developmental arrest of *hsf-1(sy441)* mutants at 25 °C. (Figure 5D). Furthermore, we found that *hsf-1(R145A)::gfp*, which has a mutation in the DNA binding site and does not form foci in response to heat stress, could not alleviate the developmental arrest phenotype of *hsf-1(sy441)* mutants at 25°C, which is in agreeance with the previous work (Figure 5 d). However, we did find that both native *hsf-1::gfp* and *hsf-1(R145A)::gfp* could rescue CID in *hsf-1(sy441)* mutants. This result suggests that the transcriptional activity of HSF-1 is required for heat-stress resistance, but not for cold adaptation. The HSF-1(sy441) protein lacks the transactivation domain of HSF-1, suggesting that this domain may play an important role in the regulation of CID. We also found that a gain-of-function mutation in *hsp-90 (p673)*, which increases lifespan in response to cold (15°C, PLoS Genet M Horikawa 2015), inhibits CID in the *hsf-1(sy441)* mutant. A previous study reported that HSP-90 forms a complex with HSF-1 and HSP-70 (G. E. Karagoz, Trends Biochem Sci. 2015). Taken together, we suggest that the interaction among HSP-70, HSP-90 and HSF-1 regulates cold acclimation.

5) The *Hsf-1(sy441)* mutation is an early stop codon which terminates the protein prior to its transactivating domain. Nevertheless, the resulting truncated *HSF-1* is still functional, and may represent a gain of function state (PMID: 25324391). Can CID be similarly achieved with *hsf-1* RNAi treatment?

➤ Thank you for your valuable comment. We examined *hsf-1* null using the *hsf-1::degron* strain as opposed to *hsf-1* RNAi and found that HSF-1 degradation led to L1 arrest at 9 and 20°C, as previously reported (Edwards SL, et al. Cell Rep. 2021, data not shown). Thus, CID is the characteristic phenotype of *hsf-1(sy441)* mutants and is not observed in the *hsf-1* null condition. But we did not examine the role of truncated HSF-1 in *hsf-1* null. Moreover, we demonstrated in Fig 5c and d that the transcriptional activity of HSF-1 may not be required for the inhibition of CID entry in *hsf-1(sy441)* mutants. This suggests that the transactivation domain, which is removed in the HSF-1(sy441) protein, may have a non-transcriptional role in cold acclimation. However, further investigation is required.

6) The authors mention several times that CID is food independent. While food is obviously present ? are you sure that enough food enters the animals?

➤ Thank you for your valuable comment. In these CID induction experiments, animals were maintained on a bacterial diet prepared according to a standard protocol (Stiernagle, T., WormBook 2006). We confirmed that wild-type animals could develop into reproductive adults under these dietary conditions in all experiments (e.g. Fig 1b). At 9°C, feeding behavior was also observed in *hsf-1(sy441)* mutants (data not shown). Therefore, *hsf-1(sy441)* mutants did consume food but we did not measure the exact food intake amount.

7) The lack of CID suppression by *daf-2* mutation, while effectively suppressing the phenotype by DAF-16 OE is contradictory and surprising. Any speculations to this contradiction?

➤ We agree with your comment. We have explained the discrepancy between overexpression of *daf-16* and the *daf-2* mutation on page 14, lines 472-476. “A previous study reported that overexpression of *daf-16* extends lifespan as well as *daf-2(e1370)* mutants, but does not induce dauer formation at 25 °C⁷¹. Thus, we suggest that the functional difference between *daf-16* overexpression and the *daf-2(e1370)* mutation on development also alters its role in CID entry.”

8) For the Putative CID suppressive mutations identified in the EMS screen ? I would expect to confirm the relevance of all of the candidate mutations in CID and/or longevity using independent mutants (generated by CRISPR, or available from other sources), and to be outcrossed several times to remove background mutations. Without showing that the CID suppression and longevity phenotype stem from the same gene ? it is impossible to argue correlation/enrichment.

- Thank you for your valuable comment. The *sur-2(ku9)* and *smg-1(cc546)* mutants are allelicly distinct from *sur-2(hru33)* in EMS #30 and *smg-1(hru112)* in #39 (Figure 8 and Extended data figure. 9). The *sur-2(ku9)* and *smg-1(cc546)* mutants were outcrossed at least four times before use, and we found that the *sur-2(ku9)* mutation suppressed CID entry and extended lifespan in wild-type animals. Other putative CID regulatory genes, namely *nsy-1*, *mtk-1*, *gst-13* and *smg-2*, are still under investigation.

9) [Redacted]

- [Redacted]

10) Smg-1 and smg-2 are part of the NMD pathway. *Hsf-1(sy441)* carries a premature stop codon and thus is a potential substrate of the NMD pathway. One must verify that the suppression by smg-1/2 is not simply due to stabilization of the *hsf-1(sy441)* transcript.

- Thank you for your valuable comment. We cannot say for sure whether *smg-1* regulates the translation of the HSF-1 protein through the NMD pathway in this work. However, previous studies have reported that *smg-1* not only activates NMD but also upregulates the UPR^{ER} chaperone gene (Masse, I. PLoS ONE 2008 and Sakaki, K. Proc. Natl. Acad. Sci. 2012). Furthermore, another study demonstrated that the UPR^{ER} chaperone gene is also upregulated through the neuronal activation of *xbp-1s*, which was used in our study in Figure 6a (Taylor and Dillin, Cell 2013.). Therefore, we believe that *smg-1* may inhibit CID through the activation of the UPR^{ER} pathway.

11) The authors think *rcd-1* might be a master regulator of CID. If so, shouldn't it be characterized more? Does it act upstream/downstream/in parallel to *HSF-1*?

- Thank you for your valuable comment. To investigate the genetic interaction between *hsf-1* genes and putative CID regulator genes, we focused on *rcd-1*, which requires CID in *hsf-1(sy441)* mutants, and generated a *Prcd-1::rcd-1::GFP* transgenic strain. However, we found that the gene expressions and patterns of *rcd-1* were not altered by environmental temperatures and the mutation of *hsf-1(sy441)* (Extended data Fig 11). We also crossed mutants of putative CID regulator genes, including *rcd-1(hru150)*, with *Phsp-4::gfp*, one of the UPR chaperones, and *Phsp16.2::gfp*, a major target gene of HSF-1, transgenic strains, but did not observe any changes in the gene expression of these strains after cold exposure (Data not shown). Therefore, we are now investigating epistasis of CID regulator genes.

Minor comments:

1) All aging-related genes tested may also be defined as stress-related genes. One should be more careful when associating them with one of these linked phenotypes.

➤ Thank you for your suggestion. In future studies, we plan to investigate whether CID is regulated by longevity machinery or stress resistance machinery.

2) Page 6 lines 180-181 - the names of the tissue specific promoters were switched.

➤ Thank you for your valuable comment. We have revised the sentences on page 8, line 286- page 9, line 289.

3) Reference 33 seems to be the wrong reference

➤ Thank you for the comment. We replaced reference 33 with Taylor and Dillin, Cell 2013.

4) Page 6 line 215 ? the *skn-1* OE result seems out of place

➤ Thank you for the comment. The result is now displayed at the right end of Extended Figure 7.

5) It is unclear how specific neuropeptides/neurotransmitters were selected for analysis.

➤ Thank you for your valuable comment. The *C. elegans* genome contains more than 100 neuropeptide precursor genes, making it challenging to test them all. Therefore, we focused on neuropeptides that are involved in cold responses, not only in *C. elegans* but also in other model organisms. We found that neuromedin U signaling plays an important role in cold-induced reproductive dormancy in *Drosophila*; therefore, decided to examine its role in CID in *C. elegans*. Previous studies have demonstrated that serotonin, one of the monoamine neurotransmitters, is involved in dauer diapause (N Fielenbach and A Antebi Genes Dev. 2008). Therefore, we focused on the functions of monoamine neurotransmitters in CID regulation. This has been mentioned in the manuscript on page 7, lines 243-246, and page 7, line 249-page 8, line 257.

Reviewer #3 (Remarks to the Author):

In this study, Horikawa et al, demonstrate that cold-inducible diapause (CID), which is induced by low temperature at 9 degrees celsius is regulated by *HSF-1*. Interestingly, a normally short-lived mutant of *HSF-1*, *hsf-1(sy441)* survives longer in this state of diapause. The authors used extensive epistasis analysis along with a forward genetic screen to identify regulators of this response. They identify several, including transcription factors required for proteostasis, as well as a neuron-to-intestine signaling pathway that via tyraminerpic signaling seems to be important for this response. While this paper is very interesting and intriguing, it also appears a bit preliminary. My major concern is that the take home message is muddled by too many factors and parallel signaling pathways involved in this response. After a lot of epistasis analysis it is still not clear how this is regulated or if there is a core pathway that regulates this from the neurons to the intestine. A lot of conclusions they are making are not really supported by the evidence (please see major questions below).

What I am also missing is a more extended Discussion, where I would have expected at least some mention of the implications for this. The authors mention the requirement for neuromedin U signaling for cold acclimation and thermogenesis in vertebrates, without further expanding on its importance. Perhaps the intestine is an important organ regulating thermogenesis as a fat-storage organ in *C. elegans*.

➤ We thank you for your many valuable comments.

A lot of open questions remain:

1) The authors say the heat shock response is required for this. But it seems more like the UPR (Ire-xbp-1 pathway) is required and ER resident Hsp70 (BiP) is induced rather than *HSF-1* regulated hsp-70 (C12C8.1). More analysis is required to clarify this.

➤ Thank you for your valuable comment. We performed a more detailed functional analysis of HSF-1. As a previous study reported (Elizabeth A. Morton, AgingCell. 2013), HSF-1 is inactive at 20 °C. This suggests that HSF-1 is not further downregulated by cold temperatures. This previous study also reported that trimerization of HSF-1 is required for heat-stress resistance and we demonstrated that native *hsf-1* alleviates developmental arrest of *hsf-1(sy441)* mutants at 25 °C. (figure 5D). Furthermore, we found that *hsf-1(R145A)::gfp*, which has a mutation in the DNA binding site and does not form foci in response to heat stress, could not alleviate the developmental arrest phenotype of *hsf-1(sy441)* mutants at 25°C, which is in agreeance with the previous work (Figure 5 d). However, we did find that both native *hsf-*

1::gfp and *hsf-1(R145A)::gfp* could rescue CID in *hsf-1(sy441)* mutants. This result suggests that the transcriptional activity of HSF-1 is required for heat-stress resistance, but not for cold adaptation. The HSF-1(sy441) protein lacks the transactivation domain of HSF-1, suggesting that this domain may play an important role in the regulation of CID. We also found that a gain-of-function mutation in *hsp-90 (p673)*, which increases lifespan in response to cold (15°C, PLoS Genet M Horikawa 2015), inhibits CID in the *hsf-1(sy441)* mutant. A previous study reported that HSP-90 forms a complex with HSF-1 and HSP-70 (G. E. Karagoz, Trends Biochem Sci. 2015). Taken together, we suggest that the interaction among HSP-70, HSP-90 and HSF-1 regulates cold acclimation.

2) The *daf-2* mutant does not rescue the CID phenotype of *hsf-1* mutant and *daf-2* mutants also do not enter CID. The authors suggest that *daf-16* overexpression rescues this. If not regulated via the DAF-2 signaling pathway, how is *daf-16* then a regulator of CID?

➤ We agree with your comment. We have explained the discrepancy between overexpression of *daf-16* and the *daf-2* mutation on page 14, lines 472-476. “A previous study reported that overexpression of *daf-16* extends lifespan as well as *daf-2(e1370)* mutants, but does not induce dauer formation at 25 °C⁷¹. Thus, we suggest that the functional difference between *daf-16* overexpression and the *daf-2(e1370)* mutation on development also alters its role in CID entry.”

3) **[Redacted]**

➤ **[Redacted]**

4) If *rcd-1* is a master regulator of CID, an epistasis analysis of other genes highlighted in this report is appropriate. For example, with the mentioned heat shock proteins or down-stream transcription factors.

➤ Thank you for your valuable comment. According to wormbase, *rcd-1/T24E12.5* is a membrane protein and may not be a transcription factor. Therefore, we will refer to *rcd-1* as a key regulator of CID rather than a master regulator. To investigate the function of *rcd-1*, we generated a *Prcd-1::rcd-1::GFP* transgenic strain (Extended data Figure 12) and investigated the genetic interaction with the *hsf-1* gene. However, the gene expressions and patterns of *rcd-1* were not altered by environmental temperatures and the mutation of *hsf-1*. These results suggest that *rcd-1* plays an important role in the regulation of CID, but does not regulate CID

through its gene expression. Further research on *rcd-1* is required, especially at the level of protein function.

5) Where is *nmur-1*, *rcd-1* and *sur-1* expressed? This would be relevant to highlight importance of the neuron-to-intestine signaling response identified here.

➤ Thank you for your valuable comment. As we specified in the response above (4), the *rcd-1* gene is expressed in the larval intestine. We did not examine *sur-1*. The *sur-2* gene is expressed in vulval precursor cells and the germ line (Singh N et al. (1995) Genes Dev, Han S et al. (2017) Nature and Grun D et al. (2014) Cell Rep). *nmur-1* is expressed in several neurons, AFD, ADF, ADL, AIA, AIZ, and RIC (Watteyne J et al. (2020) Nat Commun). AFD is a thermosensory neuron and AIZ is an interneuron involved in thermotaxis (Mori I, Ohshima Y. (1995) Nature). A tyramine synthesis gene *tdc-1* is expressed in the RIC neuron (Alkema, M.J. et al. (2005) Neuron), suggesting that neuromedin U signaling may regulate CID entry via the regulation of tyramine secretion. This suggests that neuromedin U signaling regulates cold responses via thermotaxis and tyraminerpic signaling. To prove our hypothesis, further investigations are required.

6) [Redacted]

➤ [Redacted]

Minor:

Page 3, line 101: *hsp-70* is the heat-inducible Hsp70 isoform, also known as C12C8.1 in *C. elegans*. The nomenclature “*hsp-70*” used here seems wrong and should be changed to Hsp70 if talking generally about Hsp70 and its orthologues. The Hsp70 orthologue regulated by the IRE-1/XBP-1 pathway here is the ER-resident Hsp70, also known as BiP or *hsp-4* in *C. elegans*.

➤ Thank you for your valuable comment. We have revised the sentences as per your suggestion on page 5, lines 150-152.

Page 6, line 181 and 182: pan-neuronal expression of *hsf-1* should be *unc-14p::hsf-1* and intestinal expression of *hsf-1* should be *ges-1p::hsf-1*. This has been switched here.

➤ Thank you for your valuable comment. We have revised these sentences on page 8, line 286-

page 9, line 289.

REVIEWER COMMENTS

Reviewer #1 (Remarks to the Author):

In this revision, the authors have clarified that CID is a normal physiological response to cold and showed that hsf-1 regulates the temperature sensitivity of CID. They have also addressed the minor issues from the last review.

However, the second major point on relationships among various suppressors of hsf-1 and with hsf-1 have not been addressed. It remains unclear as to whether the newly identified genes suppress hsf-1 by acting in the same or parallel pathways. Nor is there evidence to show how these suppressors interact functionally and/or regulate one another.

In the rebuttal, the authors mentioned that they tested the expression of hsf-1 targets in unnamed CID mutants and did not observe any phenotypes. However, they also show that the transcriptional activity of hsf-1 is required for regulating CID. This result indicates that testing the transcriptional targets of hsf-1 may not be the best assay for assessing regulatory interactions.

Reviewer #2 (Remarks to the Author):

In their revision, Makoto Horikawa et al improved their manuscript. Nevertheless some critical experiments are still missing (detailed below). Furthermore, I am still not convinced that CID observed in WT animals and that observed in hsf-1(sy441) mutants are the same. Some of the experiments added by the authors for the reviewers reinforce this concern. Unfortunately the author did not provide this data to the reviewers (data not shown) and did not mention them in the manuscript itself. This must be corrected, and the conclusions should be in line with this data.

Additional experiments already included in the revised version include:

- 1) an experiment in which WT animals enter a cold-induced diapaused state in *C. elegans* (CID) at 4 degrees at all developmental stages (L1-L4) (Fig2A)
- 2) Improved characterization of CID compared to other diapause states in *C. elegans* in terms of starvation, timing and genetic comparison (ED Fig 12, Fig 3C).

3) Improved functional analysis of HSF-1 showing that truncated hsf-1 CID can be suppressed by full length HSF-1 independently of its DNA binding domain (Fig 5D). This finding is surprising, and I think it is the first example of a non-transcriptional function of HSF-1. However, given the presence of several HSF-1 forms in these experiments (transactivation deficient HSF-1 along with DBD defective HSF-1) and their ability to interact with each other – the authors should be more careful when drawing conclusions and verify that indeed no DNA binding of the full length HSF-1 occurs indirectly).

4) Use of allelically distinct mutants/RNAi to verify hsf-1-associated CID suppression by sur-2, smg-1 (dig 8 and ED Fig 9).

5) Gene and expression level analysis of an rcd-1 transgene. ED Fig.11.

Additional major clarifications in the body of the manuscript are required (in addition to what has been mentioned above):

1) Characterization of the molecular and physiological hallmarks of the CID state itself (other than entry to diapause and recovery potential and P cell migration) is still required. Such CID markers (for example mitochondria status, ROS status, protein aggregation status, alae formation) should then be used to compare hsf-1-associated CID and WT-CID to see if the two are indeed similar. This is important especially now that the identified CID suppressors turned out to be hsf-1 CID-specific (mentioned in response to reviewer 1).

2) The author has also done some important experiments discussed in the response to the reviewers, but were not included in the final manuscript, nor was the data provided for the reviewers. These should be included in the main manuscript and their implication discussed. These include experiments showing that hsf-1-associated CID suppressors do not suppress CID in WT animals (mentioned in response to reviewer 1). This is an important result suggesting that either hsf-1 CID and WT CID are distinct phenomena or that the identified suppressors happen to be hsf-1-specific. This conclusion should be stated in the manuscript and in the abstract and the data should be provided.

3) Another important experiment discussed in the response to the reviewers, but were not included in the final manuscript, nor was the data provided for the reviewers is the experiment using an hsf-1 AID degron system that failed to enter CID. This result also supports the conclusion that CID entry may be specifically associated with the trans activation deficient HSF-1 form and not with HSF-1 deficiency itself. Based on this result the author should be careful not to refer to the hsf-1(sy441) strain as a loss of function (in the abstract as well as throughout the manuscript).

4) The concern that smg-1/2 suppression may be related to direct effects on NMD on the expression of the truncated hsf-1 is important, as it may indicate that smg-1/2 is or not a general part of the CID pathway. This has not been experimentally assessed, although it can be easily examined by qRT-PCR. This analysis is essential and should be provided and discussed within the manuscript.

5) Page 13 lines 440-441 discuss implication of the ire-1/xbp-1 pathway in CID. This discussion should also include the lack of requirement of xbp-1 to CID based on the xbp-1(zc12) mutant analysis.

6) RCD-1 OE affect of CID and possible epistasis should be analyzed (given the lack of changes at the expression level).

7) Was the gene name rcd-1 approved by wormbase curators?

Reviewer #3 (Remarks to the Author):

The authors have overall addressed all comments and their additional experiments have increased the clarity of the work.

I want to note that some minor points still need clarification:

line 165: please emphasize that the HSP70 orthologs you studied are ER-resident HSP70 orthologs. Otherwise this is misleading, because other cytosolic Hsp70 orthologues, which are not discussed in this study, are actually regulated by HSF-1.

Line 449-451: this sentence is overgeneralizing and implies that you have evidence that cytosolic Hsp70 and Hsp90 are important for the cold adaptation and CID entry. while this may be the case for cytosolic Hsp90 and HSF-1, you don't have proof that is the case for cytosolic Hsp70. You have shown this only for ER-resident Hsp70 orthologues, not cytosolic Hsp70 orthologues. I suggest to clarify this statement and also delete HSP70 from the sentence in lines 449-451.

REVIEWER COMMENTS

Reviewer #1 (Remarks to the Author):

In this revision, the authors have clarified that CID is a normal physiological response to cold and showed that *hsf-1* regulates the temperature sensitivity of CID. They have also addressed the minor issues from the last review.

However, the second major point on relationships among various suppressors of *hsf-1* and with *hsf-1* have not been addressed. It remains unclear as to whether the newly identified genes suppress *hsf-1* by acting in the same or parallel pathways. Nor is there evidence to show how these suppressors interact functionally and/or regulate one another.

In the rebuttal, the authors mentioned that they tested the expression of *hsf-1* targets in unnamed CID mutants and did not observe any phenotypes. However, they also show that the transcriptional activity of *hsf-1* is required for regulating CID. This result indicates that testing the transcriptional targets of *hsf-1* may not be the best assay for assessing regulatory interactions.

- Thank you for your suggestion. We agree with your concern that functional interactions between CID regulator genes and *hsf-1* remain unclear. We are still analysing the function of CID regulators but have not found key regulators of CID such as *daf-12* in dauer diapause as well as physiological hallmarks of CID that are available to monitor CID entry. To answer these questions, we need to obtain another CID mutant that has a different regulatory mechanism from the CID of the *hsf-1(sy441)* mutant and perform a comparative analysis using several CID mutants, as was done in the diapause studies.

Reviewer #2 (Remarks to the Author):

In their revision, Makoto Horikawa et al improved their manuscript. Nevertheless some critical experiments are still missing (detailed below). Furthermore, I am still not convinced that CID observed in WT animals and that observed in *hsf-1(sy441)* mutants are the same. Some of the experiments added by the authors for the reviewers reinforce this concern. Unfortunately the author did not provide this data to the reviewers (data not shown) and did not mention them in the manuscript itself. This must be corrected, and the conclusions should be in line with this data.

- Thank you for your suggestion. In this revision, we have added data that were not displayed

in the previous version of the manuscript. Please see the responses to each comment below.

Additional experiments already included in the revised version include:

- 1) an experiment in which WT animals enter a cold-induced diapaused state in *C. elegans* (CID) at 4 degrees at all developmental stages (L1-L4) (Fig2A)
- 2) Improved characterization of CID compared to other diapause states in *C. elegans* in terms of starvation, timing and genetic comparison (ED Fig 12, Fig 3C).
- 3) Improved functional analysis of HSF-1 showing that truncated hsf-1 CID can be suppressed by full length HSF-1 independently of its DNA binding domain (Fig 5D). This finding is surprising, and I think it is the first example of a non-transcriptional function of HSF-1. However, given the presence of several HSF-1 forms in these experiments (transactivation deficient HSF-1 along with DBD defective HSF-1) and their ability to interact with each other – the authors should be more careful when drawing conclusions and verify that indeed no DNA binding of the full length HSF-1 occurs indirectly).
- 4) Use of allelically distinct mutants/RNAi to verify hsf-1-associated CID suppression by sur-2, smg-1 (dig 8 and ED Fig 9).
- 5) Gene and expression level analysis of an rcd-1 transgene. ED Fig.11.

Additional major clarifications in the body of the manuscript are required (in addition to what has been mentioned above):

- 1) Characterization of the molecular and physiological hallmarks of the CID state itself (other than entry to diapause and recovery potential and P cell migration) is still required. Such CID markers (for example mitochondria status, ROS status, protein aggregation status, alae formation) should then be used to compare hsf-1-associated CID and WT-CID to see if the two are indeed similar. This is important especially now that the identified CID suppressors turned out to be hsf-1 CID-specific (mentioned in response to reviewer 1).

- We attempted to explore hallmarks of CID that can be used to perform a comparative analysis between the CID of the *hsf-1(sy441)* mutant at 9°C and the CID of the wild-type animal at 4°C. However, we have not found such features of CID yet, as described on page 13, lines 456-464. We agree with you that the identification of the hallmarks of CID is very important for thoroughly investigating its regulatory mechanism; however, we believe this to be very difficult. Because CID is a newly identified phenotype and, as we showed in the previous version of the manuscript, the regulatory mechanism of CID is parallel to dauer diapause (as described on page 12, lines 411-416). In addition, many CID regulator genes identified in this study have not been studied in other diapause phenotypes (no relevant reports were found on PubMed). This suggests

that the regulatory mechanism of CID is quite different from known diapause mechanisms. To identify the hallmarks of CID, we need to isolate another CID mutant [Redacted] and perform comparative analysis with several CID mutants, as was done in the dauer diapause study. We are now exploring other CID mutants.

2) The author has also done some important experiments discussed in the response to the reviewers, but were not included in the final manuscript, nor was the data provided for the reviewers. These should be included in the main manuscript and their implication discussed. These include experiments showing that *hsf-1*-associated CID suppressors do not suppress CID in WT animals (mentioned in response to reviewer 1). This is an important result suggesting that either *hsf-1* CID and WT CID are distinct phenomena or that the identified suppressors happen to be *hsf-1*-specific. This conclusion should be stated in the manuscript and in the abstract and the data should be provided.

➤ We have added Extended Data Fig 13 and described the result on page 12, lines 405-411. Similar to that of wild-type animals, the eggs of mutants of the CID regulator genes died completely at 4°C and did not recover from CID when the temperature was raised to 20°C. Furthermore, early L1 larvae of these mutants entered CID in the same way as wild-type animals. These results suggest that CID regulator genes are involved in CID of the *hsf-1(sy441)* mutant at 9°C (as described on page 13, lines 456-464). However, we believe that 4°C is a temperature at which *C. elegans* cannot hatch or develop; therefore, it is difficult to find any hallmark of CID at 4°C. To answer the question posed above, we are now exploring genes that can be a reporter of CID entry and will thus enable us to answer such questions. We are not able to include these conclusions in the abstract owing to word limit, but these conclusions have been stated in the discussion (on page 12, lines 405-411; page 13, lines 456-464).

3) Another important experiment discussed in the response to the reviewers, but were not included in the final manuscript, nor was the data provided for the reviewers is the experiment using an *hsf-1* AID degron system that failed to enter CID. This result also supports the conclusion that CID entry may be specifically associated with the trans activation deficient HSF-1 form and not with HSF-1 deficiency itself. Based on this result the author should be careful not to refer to the *hsf-1(sy441)* strain as a loss of function (in the abstract as well as throughout the manuscript).

➤ We have described the *hsf-1(sy441)* mutation as a premature stop codon in *hsf-1* instead of a loss-of-function mutation in the abstract and on page 4, lines 125-128. For the results of the

AID system, we have added Extended Data Fig 6 and described the result on page 9, lines 292-299. Baird et al. (2014, Science) reported that the truncated HSF-1(*sy441*) protein retains its transcriptional activity for *hsp-16.2*. We therefore tried to mimic the *hsf-1(sy441)* mutation by partially depleting HSF-1 using the AID system with low concentrations of auxin. We found that 10 μ M of auxin caused larval arrest at the L1 stage at 20°C as previously reported (Edwards et. al. Cell Rep 2021), but 1 μ M of auxin partially suppressed development in *hsf-1::degron* animals (Extended Data Fig. 6 a-b). Therefore, we thought that 1 μ M auxin would partially deplete HSF-1::degron and investigated whether CID is induced by 1 μ M auxin at 9 °C. Although treatment with 1 μ M of auxin caused developmental arrest in *hsf-1::degron* animals at 9 and 25°C, it did not induce CID at 9 °C (Extended Data Fig. 6 c-e). These results suggest that 1 μ M of auxin partially knocks down HSF-1 but did not mimic the *hsf-1(sy441)* mutation in CID formation. This result supports the hypothesis that the CID phenotype is caused by a defect in the trans-activation domain of HSF-1 and not by a reduction in the transcriptional activity of HSF-1.

4) The concern that *smg-1/2* suppression may be related to direct effects on NMD on the expression of the truncated *hsf-1* is important, as it may indicate that *smg-1/2* or not a general part of the CID pathway. This has not been experimentally assessed, although it can be easily examined by qRT-PCR. This analysis is essential and should be provided and discussed within the manuscript.

➤ We measured the gene expression of *hsf-1* in CID or L1 larvae of N2, *hsf-1(sy441)*, *smg-1(cc546)*, and *hsf-1; smg-1* at 9 °C by qRT-PCR, as described in Extended Data Fig 11d and on page 11, lines 373-382. If *hsf-1(sy441)* mRNA is degraded by the *smg-1*-mediated NMD machinery, *hsf-1* mRNA levels should decrease in the *hsf-1(sy441)* mutant and be restored by the mutation of *smg-1(cc546)*. However, we found that the mRNA level of *hsf-1* was increased in both *hsf-1(sy441)* and *hsf-1; smg-1* mutants. The gene expression of *hsf-1* was not changed in *smg-1(cc546)* mutants. This result suggests that *hsf-1(sy441)* mRNA is not degraded by the NMD machinery (as described on pages 16, lines 546-550). However, it remains unclear whether *smg-1* regulates CID entry in the *hsf-1(sy441)* mutant via the NMD machinery or another mechanism.

5) Page 13 lines 440-441 discuss implication of the *ire-1/xbp-1* pathway in CID. This discussion should also include the lack of requirement of *xbp-1* to CID based on the *xbp-1(zc12)* mutant analysis.

- We added an image of the *xbp-1(zc12)* mutant in Extended Data Fig 1a. The *xbp-1(zc12)* mutants have no phenotype at 9 °C.

6) RCD-1 OE affect of CID and possible epistasis should be analyzed (given the lack of changes at the expression level).

- We analysed the phenotype of *rcd-1::gfp* in wild-type and *hsf-1(sy441)* mutants. As shown below, *rcd-1::gfp* inhibited CID entry in *hsf-1(sy441)* mutants. We believe that the fused gfp changed the function of RCD-1, because RCD-1 is a putative membrane protein. We have not yet generated the transgenic strain that expresses high levels of native *rcd-1*. In addition, we do not present this result in the manuscript as it may confuse the reader.

7) Was the gene name *rcd-1* approved by wormbase curators?

- Yes; *rcd-1* has been assigned to the *t24e12.5* gene by the wormbase curators (https://wormbase.org/species/c_elegans/gene/WBGene00020774)

Reviewer #3 (Remarks to the Author):

The authors have overall addressed all comments and their additional experiments have increased the clarity of the work.

I want to note that some minor points still need clarification:

line 165: please emphasize that the HSP70 orthologs you studied are ER-resident HSP70 orthologs. Otherwise this is misleading, because other cytosolic Hsp70 orthologues, which are not discussed in this study, are actually regulated by HSF-1.

Line 449-451: this sentence is overgeneralizing and implies that you have evidence that cytosolic Hsp70 and Hsp90 are important for the cold adaptation and CID entry. while this may be the case for cytosolic Hsp90 and HSF-1, you don't have proof that is the case for cytosolic Hsp70. You have shown this only for ER-resident Hsp70 orthologues, not cytosolic Hsp70 orthologues. I suggest to clarify this statement and also delete HSP70 from the sentence in lines 449-451.

- Thank you for your suggestions. Accordingly, we have replaced HSP70 with ER-resident HSP70 in lines 152, 166, and 483 and have deleted HSP70 from page14, lines 492-495.

REVIEWER COMMENTS

Reviewer #1 (Remarks to the Author):

In this revision, the authors still have not address the relationship among the suppressors of hsf-1, as requested in the first review.

The authors claim that can only be done if there is “another CID mutant that has a different regulatory mechanism from the CID of the hsf-1(sy441) mutant.”

Even though the suppressors show similar phenotypes, it is still possible to test genetic and regulatory interactions among them. First, regulatory interactions can be analyzed by testing if a mutation in one suppressor gene affects the expression of a second suppressor gene. Second, genetic interactions can be assessed by testing if two suppressor mutations lead to a stronger suppression of the hsf-1 phenotype, since most of the mutations tested only lead to partial suppression of hsf-1. For example, tdc-1, rcd-1, sur-2 all show partial suppression of hsf-1. Since some of these mutations are putative nulls, if a double mutant shows stronger suppression, it would indicate that the suppressors act in parallel at least in part. Conversely, if the double mutant does not show a stronger suppression, it would suggest that the 2 suppressors act in the same pathway.

Reviewer #2 (Remarks to the Author):

In their 2nd revision, Makoto Horikawa et al addressed most of the points raised by this reviewer. I think they have done a good job in characterizing a new phenomenon of cold induced diapause in *C. elegans*. I am still not convinced that CID observed in WT animals and that observed in hsf-1(sy441) mutants are the same. Most of the characterization was done on the hsf-1(sy441) mutants and not on WT animals, and the 2 appear to have different genetic requirements. I think it is best not to claim that the 2 are the same.

Requested corrections :

1) I think the title should be changed. The heart of the paper is the identification of a new diapause state. The connection with longevity is weak at best. Some of the suppressors of CID affected lifespan and some did not. The title as it is misleading.

2) Page 5 line 150 – hsf-1 mutants are not simply chaperone mutants

3) Page 7 – The subtitle "Neural circuits require CID entry" should be rephrased . It is the other way around – CID entry requires neuronal circuits.

4) Page 8 line 274 – the claim that the fact that OE of WT hsf-1 prevents CID in hsf-1(sy441) mutants shows that the sy441 mutation is *rof*. This is untrue. It could still be that hsf-1(sy441) induces CID by *gof* , and it is the *gof* that the OE of the WT form interferes with. In fact, the *gof* interpretation is consistent with additional data presented such as the lack of CID in the hsf-1-AID system.

5) Page 10 lines 374-375 are unclear. Please rephrase.

6) Page 12 line 398 – "to rule out" is not the right phrase here.

7) Page 12 lines 403-405 – "However, *rcd-1* does not control CID through gene

expression, and therefore, the function of the RCD-1 protein may be more important in the regulation of CID." – This sentence is also wrong. What has been shown is the CID does not control RCD-1 protein levels. Not vice versa.

Reviewer #1 (Remarks to the Author):

In this revision, the authors still have not address the relationship among the suppressors of hsf-1, as requested in the first review.

The authors claim that can only be done if there is “another CID mutant that has a different regulatory mechanism from the CID of the hsf-1(sy441) mutant.”

Even though the suppressors show similar phenotypes, it is still possible to test genetic and regulatory interactions among them. First, regulatory interactions can be analyzed by testing if a mutation in one suppressor gene affects the expression of a second suppressor gene. Second, genetic interactions can be assessed by testing if two suppressor mutations lead to a stronger suppression of the hsf-1 phenotype, since most of the mutations tested only lead to partial suppression of hsf-1. For example, *tdc-1*, *rcd-1*, *sur-2* all show partial suppression of hsf-1. Since some of these mutations are putative nulls, if a double mutant shows stronger suppression, it would indicate that the suppressors act in parallel at least in part. Conversely, if the double mutant does not show a stronger suppression, it would suggest that the 2 suppressors act in the same pathway.

Communication to Reviewer #1 from the authors, Makoto Horikawa *et. al.*

Dear Reviewer #1,

Thank you for your valuable comments. We are now conducting qPCR analysis to answer the first question and generating the triple mutants to answer the second question. However, the second question is not clear to us. We were wondering if you could explain the following part of the second question in detail.

As you wrote in the response,

“For example, *tdc-1*, *rcd-1*, *sur-2* all show partial suppression of hsf-1.”

but it is not clear to us what phenotype you mean by “partial suppression”.

In our study, we have found CID (cold-inducible diapause), which is a developmental arrest observed in *hsf-1(sy441)* mutants under a cold environment (9 °C). However, we have not yet

found any morphological or physiological features of CID. Therefore, we judged whether mutants formed CID or skipped CID (non-CID) by the appearance of later stage larvae, especially adult worms. It means that our CID formation criteria does not have partial suppression, it has only CID and complete suppression of CID (non-CID).

Based on our criteria, we claim that single mutations of *tdc-1*, *rcd-1*, and *sur-2* completely suppress CID formation in the *hsf-1(sy441)* mutant but do not fully rescue the developmental delay of the *hsf-1(sy441)* mutant. This means that the phenotypes of these double mutants, *hsf-1; tdc-1*, *hsf-1; rcd-1*, and *hsf-1; sur-2*, are a complete suppression of the CID formation of the *hsf-1(sy441)* mutant, not a partial suppression of the CID formation. We therefore believe that we will not be able to observe a stronger suppression of CID in any triple mutants suggested in your response.

“if a double mutant shows stronger suppression, it would indicate that the suppressors act in parallel at least in part.”

Because their CID formation was completely suppressed.

We apologize for not explaining our CID formation criteria in detail in the manuscript and for possibly misleading you. In the revised manuscript, we will include the criteria for CID formation.

Of course, we realised that the developmental speed of *hsf-1; tdc-1*, *hsf-1; rcd-1*, and *hsf-1; sur-2* mutants was slower than that of N2 (wild-type) and *tdc-1*, *rcd-1*, and *sur-2* single mutants. And, we think that the developmental delay of these double mutants can be rescued by introducing another mutation of CID suppressor genes, for example the triple mutants suggested in your response. However, we believe that the observation of development speed cannot be used to analyse the regulatory mechanism of CID formation. Because we have no evidence to support a functional or physiological correlation between CID formation and development speed. As we claimed above, non-CID mutants in our manuscript have a complete suppression of the CID phenotype in the *hsf-1(sy441)* mutant background with showing normal development as well as wild-type animals or developmental delay. This means that it is not possible to say that the regulatory mechanism of CID formation is correlated with development speed. Therefore, even if a triple mutant such as *hsf-1;tdc-1;rcd-1* shows normal development as well as wild-type animals, it can only prove that *tdc-1* and *rcd-1* functionally interact during development, but it cannot say anything about CID formation.

Sincerely,

Reply from Reviewer #1

Dear Assistant Professor Horikawa and Professor Mizunuma,

Thank you for your letter requesting clarification regarding the issue of partial suppression of *hsf-1*. I very much appreciate the additional explanations that you have provided regarding the criteria used to infer CID phenotype.

The letter indicated that it was not possible to perform double mutant analysis because there were no partial phenotypes, and that mutants either entered CID or skipped CID. Based on the data presented, the mutations do not result in a binary effect where the whole population are 100% CID or 100% non-CID. In *hsf-1; tdc-1*, *hsf-1; sur-2*, or *hsf-1; rcd-1*, some animals arrested at the earliest stage just like *hsf-1* (Fig. 4i, 8c, 8g). This observation raises the possibility that some of these double mutants could have arrested in CID instead of developing slowly. Thus, the results are not sufficient to support the claim that CID formation was completely suppressed.

More importantly, the central point was that the manuscript should address the functional relationship between these suppressors to show whether they worked in the same or separate pathways. If you judge that the suppression phenotype is not the best route to address this point, it would be fine to show how these suppressor gene interact using alternative approaches.

We appreciate that you have read our manuscript carefully and that you provided numerous suggestions for improvement. Once again, we thank you for answering our concerns in your response.

[Redacted]

We [Redacted] performed qRT-PCR analysis for CID regulator genes in the *hsf-1(sy441)* mutant and several non-CID mutants. However, we did not observe any changes in the gene expression of CID regulators as presented in Extended Data Fig. 15. The expression of the *capa-1* gene was slightly reduced in the *hsf-1(sy441)* mutant, but this result does not align with the observation that the *hsf-1;capa-1* double mutant is a non-CID mutant. Therefore, we believe that *capa-1* gene expression is not a physiological hallmark of CID formation. We also concluded that qRT-PCR analysis is a difficult strategy to use to study the interaction with CID regulators.

Conversely, we have not yet finished the triple mutant experiments. However, as we demonstrated in the Fig. 4g–i and Extended Data Fig. 4e–h for the *hsf-1; tdc-1* mutant, Fig. 8a–c and Extended

Data Fig. 11e and f for the *hsf-1; sur-2* mutant, and Fig 8e–h for *hsf-1; rcd-1* mutant, these mutants exhibited slow developmental phenotypes, and we believe that early larval populations of these mutants that appeared to be in diapause were in fact developing very slowly and were classified as “non-CID”. Of course, these results do not rule out the possibility that some double mutants may have arrested development at a certain larval stage, but it is not possible for us to prove that the mutants have definitely arrested development and exited from diapause, as we have not yet identified any hallmark of CID. Until we discover the morphological features or marker genes that can monitor entry and exit of CID, we can only judge the CID phenotype as arrested development shortly after cold shift (CID) or non-CID. We also believed that comparative analysis using multiple CID mutants, as we propose, is the more sophisticated strategy to investigate the relationship between CID regulator genes and to explore the downstream mechanism of CID formation.

➤ [Redacted]

2. [Redacted]

➤ [Redacted]

[Redacted]

7. [Redacted]

➤ [Redacted]

Reviewer #2 (Remarks to the Author):

In their 2nd revision, Makoto Horikawa et al addressed most of the points raised by this reviewer. I think they have done a good job in characterizing a new phenomenon of cold induced diapause in *C. elegans*. I am still not convinced that CID observed in WT animals and that observed in *hsf-1(sy441)* mutants are the same. Most of the characterization was done on the *hsf-1(sy441)* mutants and not on WT animals, and the 2 appear to have different genetic requirements. I think it is best not to claim that the 2 are the same.

➤ We appreciate that you have read our manuscript carefully and provided numerous suggestions for improvement.

Requested corrections :

1) I think the title should be changed. The heart of the paper is the identification of a new diapause state. The connection with longevity is weak at best. Some of the suppressors of CID affected lifespan and some did not. The title as it is misleading.

➤ Thank you for your suggestion. Now that our manuscript focuses more on the regulatory mechanism of CID, we have changed the title to “Regulatory mechanism of cold-inducible diapause in *Caenorhabditis elegans*”.

2) Page 5 line 150 – hsf-1 mutants are not simply chaperone mutants

➤ We have corrected the text as described on page 5, lines 153-154.

3) Page 7 – The subtitle "Neural circuits require CID entry" should be rephrased . It is the other way around – CID entry requires neuronal circuits.

➤ We have corrected the text as “Neural circuits are involved in CID entry” as described on page 7, line 239.

4) Page 8 line 274 – the claim that the fact that OE of WT hsf-1 prevents CID in hsf-1(sy441) mutants shows that the sy441 mutation is *rof*. This is untrue. It could still be that hsf-1(sy441) induces CID by *gof* , and it is the *gof* that the OE of the WT form interferes with. In fact, the *gof* interpretation is consistent with additional data presented such as the lack of CID in the hsf-1-AID system.

➤ We agree with your suggestion that it is still possible that the *sy441* mutation is a loss of function as well as a gain of function. Therefore, we described CID is triggered by the protein truncation of the *hsf-1(sy441)* mutation as described on page 8, lines 285–286.

5) Page 10 lines 374-375 are unclear. Please rephrase.

➤ We believe you are referring to page 11, and we have rephrased the text on page 11, lines 385–387.

6) Page 12 line 398 – "to rule out" is not the right phrase here.

- We have modified the texts from page 12 lines 408–417 and have, therefore, deleted this phrase.

7) Page 12 lines 403-405 – "However, *rcd-1* does not control CID through gene expression, and therefore, the function of the RCD-1 protein may be more important in the regulation of CID." – This sentence is also wrong. What has been shown is the CID does not control RCD-1 protein levels. Not vice versa. In this revision, the authors have clarified that CID is a normal physiological response to cold and showed that *hsf-1* regulates the temperature sensitivity of CID. They have also addressed the minor issues from the last review.

- As suggested by reviewer #1, we measured the gene expression of *rcd-1* by qRT-PCR. Therefore, we updated the text that you described from page 12 line 419 to page 14 line 470.

- [Redacted]

[Redacted]

9) [Redacted]

- [Redacted]

[Redacted]

- [Redacted]

3) [Redacted]

- [Redacted]

6) [Redacted]

- [Redacted]

REVIEWER COMMENTS

Reviewer #1 (Remarks to the Author):

In this revision, the authors have addressed my comments.

Reviewer #2 (Remarks to the Author):

[Redacted]

They also performed qRT-PCR experiments that did not identify any relationship at the transcript level between the CID-suppressors and the CID inducing mutations.

Although I commented in my last review that I am still not convinced that CID observed in WT animals and that observed in *hsf-1(sy441)* mutants are the same, the authors continue to treat them as one. Furthermore, the CID suppressors were not examined at all in the N2 CID background. Most of the characterization was done on the *hsf-1(sy441)* mutants **[Redacted]** and not on WT animals, and the 3 appear to have different genetic requirements. I think it is best not to claim that the 2 are the same. The analysis should apply clearly only to HSF-1-CID. This point is important as it is over-interpretation of the data.

In addition, I am also worried about allele and strain-specific phenotypes.

1) HSF-1-related CID is observed in the *sy441* mutant but not in the AID mutant (perhaps this may be explained by a GOF acquired by the truncated HSF-1 mutant form).

2) **[Redacted]**

In summary, I find the manuscript is confusing.

I feel that the solid part of the manuscript is the characterization of the truncated-HSF-1 associated phenotype, which may or may not be the same as in the wild-type strain, which was not analysed in depth.

Reviewer #1 (Remarks to the Author):

In this revision, the authors have addressed my comments.

- We appreciate that you have read our manuscript carefully and have made many suggestions for improvement.

Reviewer #2 (Remarks to the Author):

[Redacted]

They also performed qRT-PCR experiments that did not identify any relationship at the transcript level between the CID-suppressors and the CID inducing mutations.

Although I commented in my last review that I am still not convinced that CID observed in WT animals and that observed in *hsf-1(sy441)* mutants are the same, the authors continue to treat them as one. Furthermore, the CID suppressors were not examined at all in the N2 CID background. Most of the characterization was done on the *hsf-1(sy441)* mutants [Redacted] and not on WT animals, and the 3 appear to have different genetic requirements. I think it is best not to claim that the 2 are the same. The analysis should apply clearly only to HSF-1-CID. This point is important as it is over-interpretation of the data.

- [Redacted]

Second, we have clarified in the manuscript that each CID refers to *hsf-1* mutant CID or wild-type CID to avoid confusion between these two types of CID. We have also tested several CID suppressors, *smg-1*, *capa-1*, *nmur-1*, *rcd-1*, and *tdc-1* in the wild-type background in Extended Data Fig. 14 and found that wild-type CID was not inhibited by the mutations of suppressors of *hsf-1(sy441)* mutant CID. In addition, we think that we are currently unable to answer whether *hsf-1(sy441)* mutant CID and wild-type CID are the same or different. Because, as we described on page 14, lines 480-485, we think that any strains are unable to hatch and develop at 4°C and therefore it is impossible to perform the developmental analysis at 4°C as we did with the *hsf-1(sy441)* mutant CID at 9°C. To answer the question, we think that we first find the hallmarks of CID by studying the mechanism of *hsf-1(sy441)* mutant CID in depth or by comparative analysis with different CID mutants, and then test the CID hallmarks in wild-type CID at the 4°C.

In addition, I am also worried about allele and strain-specific phenotypes.

1) HSF-1-related CID is observed in the *sy441* mutant but not in the AID mutant (perhaps this may be explained by a GOF acquired by the truncated HSF-1 mutant form).

➤ The previous work (Nathan A. Baird *et al.*, Science, 2014, PMID: 25324391) reported that overexpression of the truncated HSF-1(*sy441*) protein was still able to activate thermotolerance without increasing the expression of chaperone genes such as *hsp-16.2*. This suggests, as you also suggested, that the *hsf-1(sy441)* mutant is not loss-of-function, we think that the mutant is the reduction-of-function mutant. We have shown that different types of overexpression of wild-type HSF-1 inhibited CID formation of the *hsf-1(sy441)* mutant (Figure 4 a, c and g). If the truncated HSF-1(*sy441*) protein plays a role as a gain-of-function, these overexpression of wild-type HSF-1 should also induce CID in wild-type background at 9°C, but not. Therefore, we don't agree with you that the truncated HSF-1(*sy441*) protein plays a role as a gain-of-function in CID formation.

We have also shown in Extended Data Fig 6, high concentrations of auxin treatment (1-10 μM) slowed or completely inhibited the development of the *hsf-1::degron* animals between temperatures (15 to 25°C), but the *hsf-1::degron* animals was able to develop at the same temperature range with low concentrations of auxin (250-500 nM). And we were not able to find the concentration of auxin that mimicked *hsf-1(sy441)* mutation in *hsf-1::degron* animals, such as the temperature-sensitive developmental arrest not only at low temperature (9°C) but also at high temperature (25°C). Taken together, the role of the *hsf-1(sy441)* mutation is different from the loss-of-function of *hsf-1* using AID system at both low and high temperatures. Therefore, we think that the role of the *hsf-1(sy441)* mutation in CID formation needs to be further investigated.

2) [Redacted]

➤ [Redacted]

In summary, I find the manuscript is confusing.

I feel that the solid part of the manuscript is the characterization of the truncated-HSF-1 associated phenotype, which may or may not be the same as in the wild-type strain, which was not analysed in depth.

➤ We [Redacted] clarified in the manuscript that each CID refers to *hsf-1(sy441)* mutant CID or wild-type CID to avoid confusion between these two types of CID. In addition, we need to further investigate the regulatory mechanism of CID at 9°C, not at 4°C, to find morphological,

physiological, or gene expression features of CID and then we will be able to analyze CID at 4°C.